# Distinguishing rule- and exemplar-based generalization in learning systems

## Abstract

Despite the increasing scale of datasets in machine learning, generalization to unseen regions of the data distribution remains crucial. Such extrapolation is by definition underdetermined and is dictated by a learner's inductive biases. Machine learning systems often do not share the same inductive biases as humans and, as a result, extrapolate in ways that are inconsistent with our expectations. We investigate two distinct such inductive biases: feature-level bias (differences in which features are more readily learned) and exemplar-vs-rule bias (differences in how these learned features are used for generalization). Exemplar- vs. rule-based generalization has been studied extensively in cognitive psychology, and, in this work, we present a protocol inspired by these experimental approaches for directly probing this trade-off in learning systems. The measures we propose characterize changes in extrapolation behavior when feature coverage is manipulated in a combinatorial setting. We present empirical results across a range of models and across both expository and real-world image and language domains. We demonstrate that measuring the exemplar-rule trade-off while controlling for feature-level bias provides a more complete picture of extrapolation behavior than existing formalisms. We find that most standard neural network models have a propensity towards exemplar-based extrapolation and discuss the implications of these findings for research on data augmentation, fairness, and systematic generalization.

## 1 Introduction

Extrapolation or generalization—decisions on unseen datapoints—is always underdetermined by data; which particular extrapolation behavior an algorithm exhibits is determined by the algorithm's inductive biases (Mitchell, 1980). For modern deep learning systems, these inductive biases often deviate from those in humans. When the inductive biases of ML systems are opaque, and guarantees on extrapolation are not possible—as is often the case with many modern ML systems (D'Amour et al., 2020)—we can instead turn to empirical study of the *behavior* of a system to derive principles about the system's operation. Cognitive psychology provides a rich basis for experimental designs to study the often-opaque human cognitive system via its external behavior. These can be leveraged to distinguish between competing hypotheses about a machine learning system's inductive biases in the same manner (Ritter et al., 2017b; Lake et al., 2018; Dasgupta et al., 2019).

In this paper, we draw on methods from cognitive psychology to define a protocol that teases apart the different inductive biases that go into informing how an opaque learning system extrapolates outside its training distribution. We focus in particular on combinatorial generalization for classification in the presence of spurious correlation. Our protocol goes significantly beyond existing work by controlling for various confounds. We isolate two distinct kinds of inductive bias—feature-level bias and exemplar-rule bias—that have different effects on model extrapolation. We examine these inductive biases across models in an expository points-in-a-plane setting, as well as in naturalistic image and language domains. Finally, we discuss the implications of these inductive biases and their relation to previous work on data augmentation and spurious correlation.

**Feature-level bias** measures which features a system finds *easier* or *harder* to learn. This informs which feature a system will generalize on the basis of when both features are correlated or confounded. This kind of feature-level bias has been studied extensively in human cognition (Landau et al., 1988; Hudson Kam & Newport, 2005). There has also been recent work—directly inspired by these

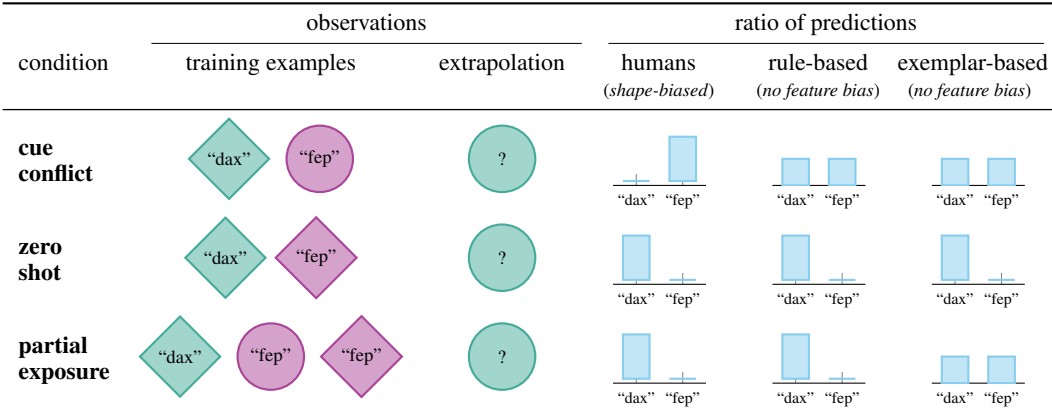

Figure 1: **Illustrative category learning experiment:** Training examples from the 3 independent training conditions, the extrapolation test, and characteristic behavior for learners with different inductive biases.

cognitive psychology studies—that examines similar biases in artificial neural networks, most notably the "shape-bias", the tendency to generalize image category labels according to shape rather than according to color or texture (Ritter et al., 2017a; Hermann et al., 2019; Geirhos et al., 2018). While there exists previous work examining specific feature biases in deep learning, we present a more general measure of feature-level bias as well as demonstrate how it interacts with—but is distinct from—another kind of inductive bias, *viz.* exemplar-vs-rule bias.

**Exemplar-vs-rule bias** measures *how* a system uses features to inform decisions by trading off between *exemplar- and rule-based generalization*. A rule-based categorization decision is made on the basis of minimal features that support the category boundary (*e.g.,* Ashby & Townsend, 1986), while an exemplar-based decision-maker generalizes a category on the basis of similarity to category exemplars (*e.g.,* Shepard & Chang, 1963), and therefore may invoke many or all features that underlie a category. Extensive empirical work in cognitive psychology has found evidence of both kinds of generalization in humans (Nosofsky et al., 1989; Rips, 1989; Allen & Brooks, 1991; Rips & Collins, 1993; Smith & Sloman, 1994). This trade-off can be understood intuitively as a continuum that varies the number of features employed to discriminate between categories (Pothos, 2005).[1] This continuum also plays a role in representation learning systems such as deep neural networks (Hinton & Salakhutdinov, 2006), where feature selection is automated.

## 2 AN ILLUSTRATIVE EXAMPLE

We first examine an intuitive example that highlights the distinct inductive biases we care about. Consider the category learning paradigm in Fig. (1). The stimuli vary along two feature dimensions, shape and color. Color determines the label of an object (*i.e.,* green objects are "dax"; purple are "fep"), and shape is unrelated to the underlying category structure and acts as a distractor. Participants (either humans or artificial learning systems) are independently placed in three different conditions— **cue conflict**, **zero shot**, and **partial exposure**—that vary in coverage of the feature space. After observing the *training examples*, the participant is presented with an *extrapolation* test consisting of an example outside the support of feature combinations observed during training (*i.e.,* to classify the green circle as a "dax" or a "fep," using arbitrary names to demonstrate that which feature is relevant to the category boundary is not given). We explain below how differences in classification behavior on this extrapolation isolate feature-level bias as well as exemplar-vs-rule bias. We encourage the reader to try the experiment themselves to examine their intuitions.

**Cue conflict** (CC, top row, Fig. (1)). The data presented in this condition confound color and shape (*i.e.,* color and shape are equally predictive of the category boundary). How a system generalizes here directly measures its feature-level bias towards color or shape.

---

[1]We leave to future work the details of mathematically formalizing the properties of statistical learners that result in exemplar-vs-rule bias. We instead focus on the behavioral manifestations of this inductive bias and present an empirical protocol to measure it, even in opaque systems.

*Characteristic behavior* (right half of Fig. (**1**)). Since humans have an established shape bias (Landau et al., 1988), we expect that humans will classify the test item according to the object that shares its shape, not its color; in this case, as a "fep." However, this inductive bias is not shared by rule- and exemplar-based reasoners that have no *a priori* propensity for features, and are equally likely to classify the test item as a "dax" or a "fep."

**Zero shot** (ZS, middle row, Fig. (**1**)). This condition requires extrapolation to a new feature value "zero-shot" (*i.e.,* without prior exposure). This setting is often used to examine out-of-domain (OOD) and compositional generalization in machine learning (Xian et al., 2018). Behavior in this condition reveals whether the model has learned the discriminating features and whether it can extrapolate to new feature values, and thus acts as a baseline.

*Characteristic behavior* (right half, Fig. (**1**)). Rule- and exemplar-based behavior in this condition is confounded. A rule-based learner infers the minimal rule that color determines label, does not assign any predictive value to shape, and therefore classifies the extrapolation stimulus based on color as a "dax." An exemplar-based learner categorizes based on the similarity along all feature dimensions of the extrapolation stimulus to category exemplars. Both training exemplars have no overlap with the test stimulus along the shape dimension, but the "dax" overlaps along the color dimension, and the learner categorizes it as a "dax."

**Partial exposure** (PE, bottom row, Fig. (**1**)). Compared to zero shot, participants in this condition also receive "partial exposure" to a new feature value (*i.e., circle*) along the shape dimension. This setting is most similar to *combinatorial zero-shot generalization* (*e.g.,* Lake & Baroni, 2018a), where the learner is exposed independently to all feature values but has to generalize to a new combination.

*Characteristic behavior* (right half of Fig. (**1**)). This setting meaningfully distinguishes rule- and exemplar-based generalization. To understand the behavior of these two systems, we contrast this condition to the cue-conflict condition. The addition of the purple diamond-shaped "fep" means the learner has seen both a diamond and a circle labeled "fep". A rule-based learner takes this as direct evidence that shape is *not* predictive of category label and classifies the extrapolation stimulus on the basis of color as a "dax." This is typically also how humans extrapolate. This additional training example, however, does not impact an exemplar-based system, since it does not share any features with the extrapolation stimulus. The exemplar-based reasoner classifies on the basis of feature-overlap with training exemplars and is therefore indifferent, exactly as in the cue-conflict condition.

**From behavior to inductive bias.** *Feature-level bias* is measured as deviation from chance performance in the CC condition. *Exemplar-vs-rule bias* is measured by the difference between performance in the PE and ZS conditions—-no difference indicates rule-based generalization, while the magnitude of the difference measures exemplar propensity. Pure exemplar-based reasoning implies no difference between the PE and CC conditions, while a non-zero difference indicates partial rule propensity.

**Implications.** A purely exemplar-based system doesn't learn decision boundaries that operate over minimal features. It instead favors a decision boundary that weights all features. This is undesirable in domains where not all feature combinations will be observed, and systematic generalization to unobserved combinations is expected (Lake et al., 2018; Marcus, 2018; Arjovsky et al., 2019). On the other hand, a rule-based system that applies the same category decision rules across all data regions might over-generalize, which is undesirable in some naturally occurring long-tailed distributions (Feldman & Zhang, 2020; Feldman, 2020; Brown et al., 2020). In such cases, flexible exemplar-based learning that generalizes based on dense similarity is preferable (Zhang et al., 2016; Arpit et al., 2017). The protocol we present allows us to empirically measure the exemplar-vs-rule bias of a learner and therefore navigate the exemplar-vs-rule trade-off in various scenarios.

## 3 A PROTOCOL FOR EXAMINING INDUCTIVE BIAS

We embed the structure of the category learning problem discussed in Section (**2**) into a statistical learning problem that can be applied across domains to test black-box learners.

**Problem setting.** We consider a setting where inputs are a composition of categorical attributes (*oracle* setting in Andreas, 2019) with two latent binary features, $\mathbf{z}_{\text{disc}}, \mathbf{z}_{\text{dist}}, \in \{0, 1\}$ that jointly determine the observation $\mathbf{x}$ via some mapping $g : \{0, 1\}^2 \to \mathcal{X}$; see Fig. (**2**). These features can be derived from a richer set, *e.g.,* the median of a continuous feature (see Appendix). We consider the

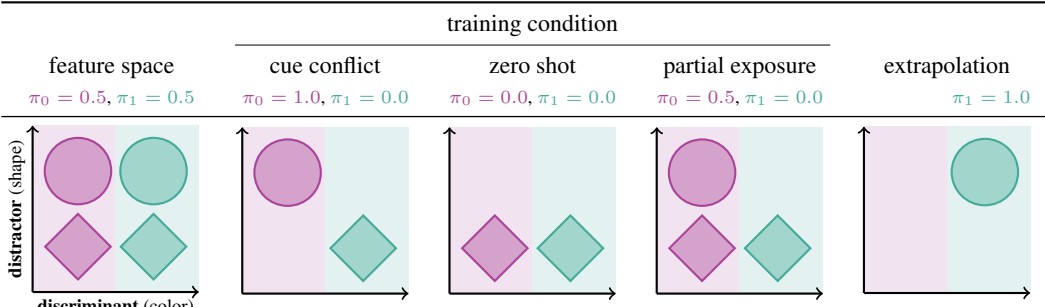

**Figure 2: Formalizing the illustrative experiment:** The experiment from Fig. (1) expressed in terms of the formalism in Section (3) with $\mathbf{z}_{\text{dist}}$ = color and $\mathbf{z}_{\text{disc}}$ = shape. Background colors indicate true category boundary.

binary classification task of fitting a model $\hat{f} : \mathcal{X} \rightarrow \{0, 1\}$ from a given model family $\mathcal{F}$ to predict a binary label for each observation. One of the underlying features, the *discriminant*, $\mathbf{z}_{\text{disc}}$, defines the decision boundary; the other one, the *distractor*, $\mathbf{z}_{\text{dist}}$, is not independently predictive of the label.

This specifies a generative process $\mathbf{x}, \mathbf{z}_{\text{disc}}, \mathbf{z}_{\text{dist}} \sim p(\mathbf{x} \mid \mathbf{z}_{\text{disc}}, \mathbf{z}_{\text{dist}}) \, p(\mathbf{z}_{\text{disc}}, \mathbf{z}_{\text{dist}})$. $p(\mathbf{x} \mid \mathbf{z}_{\text{disc}}, \mathbf{z}_{\text{dist}})$ is either generated (*e.g.,* in Section (4)), or the empirical distribution of the subset of datapoints $\mathbf{x}$ with the corresponding underlying feature values (assuming access to these annotations, *e.g.,* in Sections (5) and (6)). $p(\mathbf{z}_{\text{disc}}, \mathbf{z}_{\text{dist}})$ is varied across training conditions, as outlined below.

**Training conditions.** The upper-right quadrant in all subfigures of Fig. (2), for which $p(\mathbf{z}_{\text{disc}} = 1, \mathbf{z}_{\text{dist}} = 1) = 1$, acts as a hold-out set on which we can evaluate generalization to an unseen combination of attribute values. We produce multiple training conditions with the remaining three quadrants of data by manipulating $p(\mathbf{z}_{\text{disc}}, \mathbf{z}_{\text{dist}})$. All the analyses in this paper compare model extrapolation to the held-out test quadrant across various training conditions.

To equalize the class base rates we balance all training conditions across the discriminant; *i.e.,* we enforce $p(\mathbf{z}_{\text{disc}} = 0) = p(\mathbf{z}_{\text{disc}} = 1) = 0.5$. We also fix the number of datapoints across all conditions at $N$; With these constraints, we can control $p(\mathbf{z}_{\text{disc}}, \mathbf{z}_{\text{dist}})$ via two degrees of freedom: $\pi_0 = p(\mathbf{z}_{\text{dist}} = 1 \mid \mathbf{z}_{\text{disc}} = 0)$ (this implicitly fixes $p(\mathbf{z}_{\text{dist}} = 0 \mid \mathbf{z}_{\text{disc}} = 0) = 1 - \pi_0$ to balance the dataset); and $\pi_1 = p(\mathbf{z}_{\text{dist}} = 1 \mid \mathbf{z}_{\text{disc}} = 1)$. The three conditions in Section (2), as well as the held-out test set, correspond to particular settings of $\pi_0$ and $\pi_1$ (shown in Fig. (2), more in Appendix (A.2)).

**Measuring inductive bias.** For a given model family $\mathcal{F}$, let $\hat{f}^{\text{ZS}}$ denote the result of selecting a model from $\mathcal{F}$ by training in the zero-shot condition, and similarly $\hat{f}^{\text{PE}}$ and $\hat{f}^{\text{CC}}$. Feature-level bias (FLB) and exemplar-vs-rule propensity (EVR) are measured as:

$$\text{FLB}(\mathcal{F}) = \mathbb{E}[(A(y, \hat{f}^{\text{CC}}(\mathbf{x}))] - 0.5 \, , \tag{1}$$

$$\text{EVR}(\mathcal{F}) = \mathbb{E}[A(y, \hat{f}^{\text{ZS}}(\mathbf{x}))] - \mathbb{E}[A(y, \hat{f}^{\text{PE}}(\mathbf{x}))] \tag{2}$$

where the expectation is taken with respect to the the data distribution under the extrapolation region (*i.e.,* $p(\mathbf{x}, y \mid \pi_0 = 1, \pi_1 = 1)$), $A$ is the 0-1 accuracy. FLB takes values between -0.5 and 0.5 (indicating bias toward $\mathbf{z}_{\text{dist}}$ or $\mathbf{z}_{\text{disc}}$, respectively); 0 represents no feature bias. EVR takes values between 0 and 1 (indicating rule-based and exemplar-based extrapolation, respectively).

**Related formalisms and spurious correlation.** This binary formulation of discriminant and distractor features has previously been studied in the context of spurious correlation (Sagawa et al., 2020). Rather than independently varying occupancy in the four quadrants, they directly manipulate the (spurious) linear correlation between the distractor and the discriminant features ($p_{maj}$). In combinatorial feature spaces, a scalar spurious correlation insufficiently specifies the data distribution. The linear correlation coefficient $\rho$ between $\mathbf{z}_{\text{disc}}$ and $\mathbf{z}_{\text{dist}}$—henceforth "spurious correlation"—can be written in terms of $\pi_0$ and $\pi_1$:

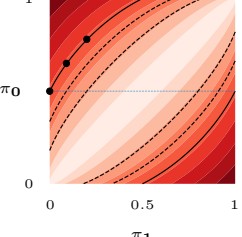

**Figure 3:** Spurious correlation across data distributions (Eq. (3)).

$$\rho(\pi_0, \pi_1) = \frac{\alpha}{\sqrt{\beta(1 - \beta)}}; \qquad \alpha = \frac{\pi_0 - \pi_1}{2}, \ \ \beta = \frac{\pi_0 + \pi_1}{2} \, . \tag{3}$$

Different $\pi_0$ and $\pi_1$ combinations can give equal $\rho$ (see contours in Fig. (3), dots indicate points along the equi-correlation contour that intersects with the PE condition: ($\pi_0 = 0.5, \pi_1 = 0.0, \rho = 0.58$)). In this paper we demonstrate that different data distributions with the same spurious correlation can result in vastly different generalization behavior to the under-represented extrapolation quadrant. This indicates that sensitivity to spurious correlation is an underspecified inductive bias—we risk conflating conceptually distinct sources of inductive bias by focusing on this single metric. We argue for a formulation like ours—based on manipulating feature *combinations*—that can tease apart distinct inductive biases: at the level of what features a system finds easier to learn (FLB) as well as how to use these features to inform a decision boundary (EVR). We discuss how these biases can be interdependent, but capture distinct behaviors that sensitivity to spurious correlation cannot explain, thereby providing a more comprehensive picture of how a system generalizes.

## 4    EXEMPLAR-VS-RULE BIAS IN A 2-D CLASSIFICATION EXAMPLE

To illustrate our framework in a simple statistical learning problem, to quantitatively confirm the intuitions outlined in Section (2), we consider a two-dimensional classification problem. The feature dimensions are orthogonal bases in 2D space. We specify:

$$p(\mathbf{x} \mid \mathbf{z}_{\text{disc}}, \mathbf{z}_{\text{dist}}) = \mathcal{N}(\mu, 1.0); \qquad \mu = \alpha \times [2\mathbf{z}_{\text{disc}} - 1, 2\mathbf{z}_{\text{dist}} - 1] \qquad (4)$$

where, as specified in Section (3), $\mathbf{z}_{\text{disc}}, \mathbf{z}_{\text{dist}}, \in \{0, 1\}$, $p(\mathbf{z}_{\text{disc}}, \mathbf{z}_{\text{dist}})$ is determined by the training condition. $\mathbf{z}_{\text{disc}}$ determines class labels, $\mathbf{z}_{\text{dist}}$ is a distractor, $\alpha$ is fixed at 3, and $N = 300$ datapoints are in each class. The group with $\mathbf{z}_{\text{dist}} = \mathbf{z}_{\text{disc}} = 1$ is assigned the test set.

### 4.1    MODEL FAMILIES AND NOMENCLATURE.

**Neural network (NN):** We train feedforward ReLU classifiers with varying numbers of hidden layers and hidden units. We use the scikit-learn implementations with default parameters, run 20 times for confidence intervals.

**Generalized linear model (GLM):** Parametric models allow us to formalize the feature-sparsity that characterizes rule-based learners. Linear logistic regression is sparse by definition (it has access to only linear features). We generalize this model by expanding the feature space to include a nonlinear interaction $\Phi$ and examine L1 and L2 regularization on in a GLM over this altered feature space.

**Gaussian process (GP):** Non-parametric kernel methods allow us to formalize exemplar-based generalization, where generalizations are made on the basis of dense similarity to training data. A direct implementation of exemplar-based reasoning is only possible in the synthetic setting in which features over which to compute similarity are known. We examine the performance of GPs with Radial Basis Function (RBF) kernels. We fit the kernel length-scale (Rasmussen, 2003) (giving 5.2) as well as vary it (adjusting "locality" in decision boundaries); GP:8.0 denotes a lengthscale 8.0 GP.

### 4.2    COMPARING CUE CONFLICT, ZERO SHOT, AND PARTIAL EXPOSURE

We consider one model from each class: NN with 1 hidden layer, 2 units (NN:2h1d); linear GLM (GLM:lin); RBF GP with fitted lengthscale (GP:fit). The decision boundaries learned by these models are shown in Fig. (4a). $\mathbf{z}_{\text{dist}}, \mathbf{z}_{\text{disc}}$ are equivalent by design, and permit no FLB, CC is exactly at chance. This lets us focus on validating our novel protocol for measuring EVR without confounds. We generalize to cases with FLB in later sections. The GLM, sparse and therefore rule-based by definition, can only learn a linear boundary. It is therefore unaffected by the distractor dimension, showing no difference in extrapolation behavior between ZS and PE (zero EVR). On the other hand, the GP is exemplar-based by definition and displays a high EVR. The NN shows an intermediate EVR, more rule-based than the purely-exemplar-based GP but not entirely rule-based like the GLM.

### 4.3    THE INFLUENCE OF MODEL PROPERTIES ON THE EVR

We first examine EVR in our control model classes (GLMs and GPs) to validate that it tracks rule- vs exemplar-based extrapolation, followed by analyses of various NNs.

**Regularized GLMS: EVR reduces with rule propensity.** A key property of rule propensity is sparsity in feature space. A linear GLM (GLM:lin) is sparse by definition, we examine a GLM on an

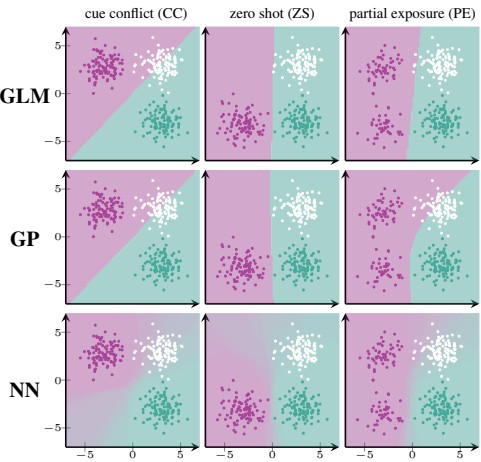
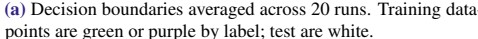

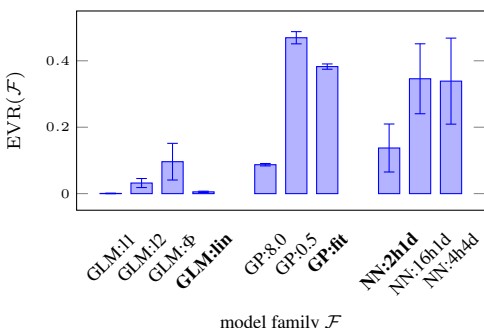

(b) **EVR reflects exemplar-vs-rule propensity both within and across model families.** The EVR across model families, computed across 20 runs, error bars represent 95% CIs. The GLMs are largely rule-based and show low EVR. Even within GLMs, sparsity regularization gives lower EVR. GPs are largely exemplar-based and show high EVR. Even within GPs, more 'local' GPs with lower lengthscales have higher EVR. NNs lie in-between, with larger NNs giving higher EVRs.

(a) Decision boundaries averaged across 20 runs. Training datapoints are green or purple by label; test are white.

**Figure 4:** **Simple 2-D classification** (Section (4)) The specific model used in (a) are bolded in (b).

expanded feature set so we can manipulate this sparsity. The additional feature $\Phi \propto \mathbf{z}_{\text{dist}} * \mathbf{z}_{\text{disc}}$ is computed by taking the product of the observed features and normalizing by $\alpha$. We compute EVR for this GLM with different regularizers (regularization weight 1.0), shown in Fig. (4b).

GLM with no regularization (GLM:$\Phi$) displays a significant EVR. L2 regularization reduces it but L1 (which directly induces feature sparsity[2]) brings it to zero (or perfectly rule-based). This shows that a low EVR tracks rule propensity *i.e.,* feature-level sparsity in a model.

**Lengthscales in GPs: EVR increases with exemplar propensity.** A sufficient condition for exemplar propensity is the locality of decision boundaries. We can directly manipulate this in a GP with its lengthscale. We evaluate EVR in GPs with RBF kernels of different lengthscales in Fig. (4b). We find that the EVR is lowest with high length-scales and grows as the lengthscale reduces. This shows that a high EVR tracks exemplar propensity, *i.e.,* having local decision boundaries.

**NNs: The necessary but not sufficient role of expressivity.** The results from GLMs and GPs above indicate that some ways to reduce expressivity (L1 regularization in GLMs and high lengthscale in RBF-GPs) encourage rule propensity over exemplar propensity (thereby a lower EVR). We manipulate the most common variable in NN expressivity—its size.

We increase the width of an NN with fixed depth of 1 (Fig. (4b)) and find that the EVR increases. A deep NN with the same number of units, however, shows a comparable EVR to a wide network. Deeper networks with the same number of units are more expressive than wide ones (Raghu et al., 2017), indicating that excess expressivity—while necessary—is not the sole driver of EVR.

### 4.4 EVR IS DISTINCT FROM SENSITIVITY TO SPURIOUS CORRELATION

A crucial difference between the ZS and the PE conditions is that the PE condition creates a (spurious) correlation $\rho = 0.58$ between $\mathbf{z}_{\text{dist}}$ and $\mathbf{z}_{\text{disc}}$. Is sensitivity to this spurious correlation ($\rho$) the sole the driver of the difference in performances between the PE and ZS conditions, *i.e.,* of the EVR? We show that this is not the case; the EVR is measuring something distinct. As described in Section (3), there are multiple data-settings with the same $\rho$. We consider training conditions specified by other $\pi_0, \pi_1$ that have the same $\rho$ as the PE condition (dots along the solid contour in Fig. (3)). We find that performance on the extrapolation quadrant after training on these new data distributions is much higher (and closer to ZS performance) than when trained on the PE condition—even though $\rho$ is exactly the same. This indicates that performance on the PE condition (normalized by ZS performance to give the EVR) is uniquely indicative of something different from sensitivity to spurious correlations—it measures the inductive bias toward exemplar-vs-rule based extrapolation.

---

[2]Weight sparsity from L1-regularizer is equivalent to feature-sparsity only in special cases, including GLM.

We also find that different ways to reduce $\rho$ give different extrapolation behavior (*e.g.,* increasing $\pi_1$ is more effective than reducing $\pi_0$, see Appendix for details). This has implications for data manipulation methods (*e.g.,* subsampling or augmentation) that manipulate this $\rho$ to control extrapolation. This also supports that spurious correlation alone cannot explain extrapolation behavior, highlighting the importance of FLB and EVR that measure behavior under different feature *combinations* in training.

**Conclusions.** The EVR tracks exemplar-vs rule-based extrapolation, as validated by evaluating it on interpretable models like GLMS and GPs. It decreases with reductions in expressivity mediated by regularization and lengthscale. The EVR in NNs also decreases with (some kinds of) expressivity: it is more sensitive to increases in width than depth. Finally, sensitivity to spurious correlation cannot explain the EVR. These results lay groundwork for future theoretical work in formalizing the EVR.

## 5 CREATING COMPOSITIONALITY BY ADDING DISTRACTORS TO IMDB

In this section, we show how we can apply our approach to any classification task. We consider sentiment analysis on Internet Movie Database Movie Reviews (IMDb) (Maas et al., 2011).

**Selecting features.** The sentiment label ("positive" or "negative") is the discriminant $\mathbf{z}_{\text{disc}}$. We manufacture an orthogonal distractor $\mathbf{z}_{\text{dist}}$ as the presence/absence of a word that a) occurs in roughly $50\%$ of the sentences in the dataset and b) does not occur more frequently for either positive or negative reviews. Some examples are "film" and "you": we use the word "film," see Figure 5.

**Models.** We train a single layer LSTM (20 hidden units; default hyperparameters) on each condition and test on the held-out quadrant. We exclude models that do not reach $80\%$ validation accuracy.

**Feature-level bias.** The distractor $\mathbf{z}_{\text{dist}}$ is easier to learn than the discriminant $\mathbf{z}_{\text{disc}}$, as reflected in the CC condition (19.7%, FLB = $-0.3$): the system preferentially classifies on the basis of the distractor.

**Exemplar-rule bias.** We see good performance in ZS (84%): despite never having seen the word "film," the system can generalize to reviews containing it. The performance in PE drops significantly (30.1%) giving a large EVR (EVR = 0.54), indicating exemplar-based reasoning. Note that the PE condition has access to a superset of feature coverage compared to ZS, and contains all the same information required to generalize well. However, the exemplar-based tendency to nonethe-

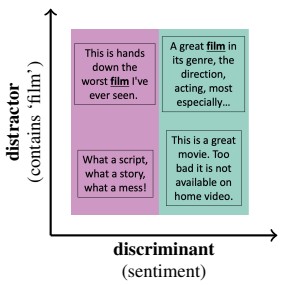

**Figure 5:** Example stimuli from the IMDb dataset.

less index on additional features (*e.g.,* the presence of the word "film") hurts performance on the extrapolation quadrant. Performance in PE is higher than in CC, indicating that the system can learn to use the discriminant (*i.e.,* it is not purely relying on FLB).

## 6 A COMPOSITIONAL DATASET DERIVED FROM MULTI-LABEL CELEBA

We now test our protocol on a standard classification task on a large-scale image dataset, CelebFaces Attributes (CelebA) (Liu et al., 2015). Each image in this dataset is labelled with 40 binary attributes, each of which can be leveraged as discriminant or distractor. We examine FLB and EVR for standard models across different feature pairs, and discuss the practical implications of our findings.

**Selecting features.** We select feature pairs that split the data roughly evenly and thus maximizing the number of training datapoints in each quadrant. We carry out our analyses across a range of feature pairs; an example is depicted in Fig. (6a), and further details are in the Appendix.

**Models.** We train ResNets of various depths ($\{10, 18, 34\}$) and widths ($\{2, 4, 8, 16, 32, 64\}$) on 6 different choices for feature pairs, with standard hyperparameters. We limit our analyses to networks that achieve at least $75\%$ validation accuracy to ensure that, despite differences in data variability across training conditions, all models learn a meaningful decision boundary.

**Feature-level bias.** There is a wide range of FLB in the feature pairs; *e.g.,* "male" is easier to learn than "high cheekbones" giving high FLB, "mouth open" and "wearing lipstick" are equally difficult and give an FLB of close to 0. These FLBs were consistent across ResNet widths and depths.

**Exemplar-rule bias.** We observe good ZS performance: the models can generalize to new feature values outside the training support. We see a wide range of EVR across feature pairs, Fig. (6b). Across all feature pairs, the EVR is non-negative: generalization in the PE condition is always worse (or not significantly better) than in the ZS condition. Further, we see a linear correlation between EVR and FLB in logit space across feature pairs. EVR therefore depends on how easy or hard the features are to learn. The key, however, is that this regression of the EVR onto FLB has a positive intercept: there is a positive EVR even for feature pairs with no FLB. There is a tendency for lower performance in PE compared to ZS (a nonzero EVR, exemplar propensity) even when FLB is controlled for.

We find no differences in EVR across ResNet widths and depths: Fig. (6b) plots EVR and FLB averaged over ResNet sizes (model-specific results in Appendix). One explanation is that the features in CelebA are complex; to learn these, we need reasonably high model expressivity, and differences in parameter count do not further modulate EVR. This is consistent with findings in Section (4) where expressivity is necessary but not sufficient for increases in EVR: we see a jump in EVR going from NN:2h1d to NN:16h1d, but no further change going to the even more expressive NN:4h4d.

**Controlling spurious correlation.** We replicate the findings in Section (4): the EVR cannot be explained by sensitivity to spurious correlation $\rho$. This is shown in Fig. (6c), where we substitute performance in the PE condition with performance in a different data-setting ($\pi_0 = 0.825, \pi_0 = 0.25$) with the same $\rho = 0.58$ as in the PE condition. We find none of the effects discussed above, indicating that the PE condition is measuring something unique—exemplar-vs-rule propensity—which is not accounted for by sensitivity to spurious correlation. Further, EVR does not increase with model expressivity, unlike sensitivity to spurious correlation (Sagawa et al., 2020).

**Practical implications of the EVR.** The nonzero EVR, *i.e.,* exemplar-basedness reveals that our models are better at extrapolating zero-shot to a new feature value than when they have partial exposure to that feature value, *even though the additional data need not change the learned decision boundary*. The training examples added in PE can be classified with the decision function from ZS without incurring additional training loss. A rule-based system recognizes this and bases its generalization on the minimal features that support the category boundary, *i.e.,* the ones also learned in the zero-shot case. However, an exemplar-based model changes its decision boundary in response to this additional data. This has implications for fairness, as we outline with example features below.

PE-approximating data distributions ($\pi_0 \approx 0.5, \pi_1 \approx 0.0$) occur naturally. For example, as Sagawa et al. (2020) observe, "blond" "male"'s are under-represented in CelebA. Consistent with the rest of our results, we find better classification for the extrapolation quadrant (blond males) if we discard data from an adjacent quadrant (blond non-males, or non-blond males) simulating the zero-shot condition, as opposed to the PE condition if such data is included: ResNet10, width 2, gives $ZS = 75.12 \pm 3.09\%$; $PE = 60.22 \pm 7.27\%$ for $\mathbf{z}_{\text{disc}} =$"male" (discard blond non-males to get ZS) and $ZS = 68.16 \pm 3.34\%$; $PE = 49.78 \pm 3.76\%$ for $\mathbf{z}_{\text{disc}} =$"blond" (discard non-blond males to get ZS). Previous sub-sampling approaches (Sagawa et al., 2020; Haixiang et al., 2017) that manipulate spurious correlation underspecify feature distributions (in the sense that there are many ways to alter feature distributions to change spurious correlation (Fig. (3))), and the aforementioned results demonstrate that these distinctions are important in determining extrapolation behavior to held-out feature combinations. Moreoever, differences in extrapolation behavior between the PE and ZS conditions, in particular, give insight into a system's exemplar-vs-rule bias.

## 7 RELATED WORK AND FUTURE DIRECTIONS

**Model design for systematic generalization.** Rule-based generalization permits systematic extrapolation in combinatorial domains. This systematicity has been found lacking in neural networks (Lake & Baroni, 2018b; Barrett et al., 2018), leading to renewed interest in hybrid symbolic–connectionist architectures (Garnelo & Shanahan, 2019). However, works evaluating, or proposing, new algorithms for systematic generalization have not performed a thorough investigation of how feature co-occurrences modulate extrapolation behaviors. Evaluating the exemplar-rule trade-off in these models by evaluating their EVR is a promising future direction.

**Learning causal features.** Feature sparsity, as exemplified by rule-based generalization, is equivalent to learning causal features under the assumption that the causal model is the simplest model that explains the data. Recent work has investigated data settings that permit the separation of causal

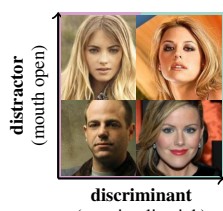

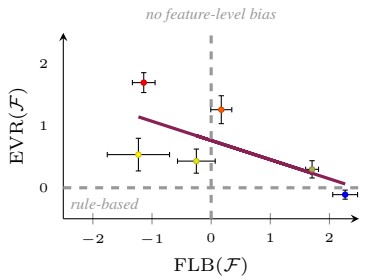

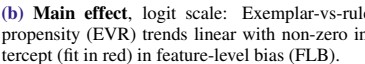

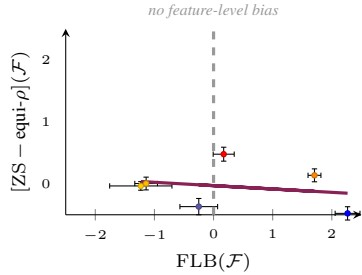

(a) Example CelebA stimuli; we test 6 discriminant-distractor pairs (with sufficient coverage of quadrants in Fig. (2)) of 6 features.

(b) **Main effect**, logit scale: Exemplar-vs-rule propensity (EVR) trends linear with non-zero intercept (fit in red) in feature-level bias (FLB).

(c) **Control**, logit scale: Performance deviation from ZS under equi-correlation interpolation is close to zero; linear fit (red) has intercept at zero.

Figure 6: **CelebA results.** Stimuli and results on various feature pairings from the CelebA domain (Section (6)). Error bars represent 95% confidence intervals across ResNets of various sizes. See figure sub-captions and main text for details.

features from spurious correlations (Arjovsky et al., 2019; Bengio et al., 2019; Hill et al., 2019). Here, we demonstrate that a model with an exemplar propensity makes more rule-based ("causal") extrapolation for certain training feature combinations (*i.e.,* zero shot as opposed to partial exposure). Investigating how feature coverage impacts causal generalization is a fruitful future direction.

**Similarity-based generalization and kernels.** We use similarity-based kernels such as the RBF to exemplify exemplar-based extrapolation. Recent work has interpreted neural networks as kernel regression (Jacot et al., 2018). Theoretically formalizing what NN properties, under a kernel framing, lead to being exemplar-based (as behaviorally tracked by the EVR) is an exciting future direction.

**Data augmentation.** The EVR demonstrates that increased data variation in the form of feature coverage worsens systematic generalization. The negative effect of data variation on generalization has been documented for adversarial augmentations (Raghunathan et al., 2020). Our results show that this phenomena persists even when augmentation is not adversarial (not maximizing a loss), rendering it generally relevant for the design of data augmentations (Perez & Wang, 2017).

## 8 CONCLUSIONS

Taking inspiration from—and going beyond—psychological studies, we design a behavioral protocol to distinguish the effects of two inductive biases (FLB and EvR) that is easily applicable to any classification domain. This follows in a promising line of recent work that analyses and interprets deep learning systems based on their external behavior (Ritter et al., 2017b; Dasgupta et al., 2019). It complements other approaches that follow in the neuroscience tradition of analyzing internal representations (Zeiler & Fergus, 2014; Karpathy et al., 2015), or make approximations of these internal workings to support theoretical results (Jacot et al., 2018; Li & Liang, 2018; Allen-Zhu et al., 2019; Du et al., 2018). The behavioral approach has the advantage that it makes no assumptions about the model, allowing comparisons across models that differ in complexity and architecture.

Both rule- and exemplar-based extrapolation are valuable depending on domain, underscoring the importance of diagnosing this bias in systems. We show that this inductive bias is distinct from sensitivity to spurious correlation (Sections (4) and (6)), highlighting the impact of imbalanced feature *combinations* on extrapolation in neural networks (as captured by FLB and EVR).

Finally, the EVR is also a novel empirical phenomenon. As demonstrated in two real-world domains, we find that more feature coverage (as in PE compared to ZS) hurts generalization for exemplar-based models. This has implications for methods that manipulate data distributions to improve performance; *e.g.,* data subsampling (Haixiang et al., 2017), data augmentation (Perez & Wang, 2017; Antoniou et al., 2017), and contrastive learning (Chen et al., 2020). Since an exemplar-based model is easily biased by the inclusion of a distractor feature, EVR is potentially useful as a diagnostic in application settings (*e.g.,* fairness) in which the goal is to control algorithmic behavior on non-representative factors of a dataset (*i.e.,* algorithmic bias Mitchell et al. (2019)).

A limitation of the present work is that we do not provide a conclusive answer as to what properties of a model family influence the EVR. A broader study on these controlling factors (*e.g.,* hyperparameters, optimizers), and theoretical work formalizing this effect, are exciting avenues for future work.

REPRODUCIBILITY STATEMENT

We evaluate reproducibility according to the criteria from Pineau et al. (2020) in Appendix (D). We additionally provide source code in a `zip` file as supplementary material.

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

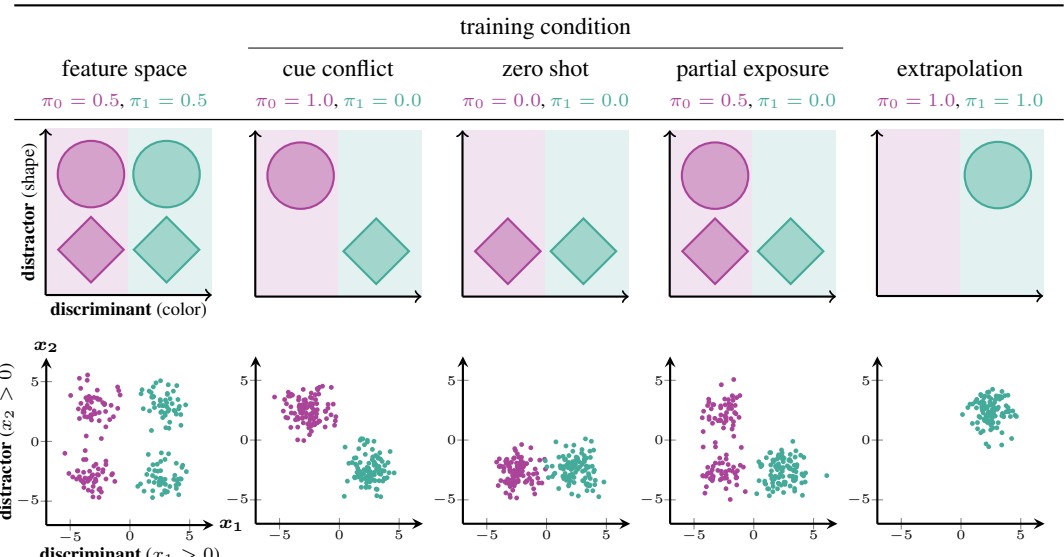

**Figure 7:** We expand on Fig. (2) from the main text by including a realization of the abstract training conditions in the simple 2D points-in-a-plane setting. (**Top**) **Formalizing the illustrative experiment:** The experiment from Fig. (1) expressed in terms of the formalism in Section (3) with $\mathbf{z}_{\text{dist}} = $ color and $\mathbf{z}_{\text{disc}} = $ shape. Background colors indicate true category boundary. (**Bottom**) The conditions realized via a binarization of continuous feature values. Here, the discriminant is binarized as $x_1 > 0$ and the distractor as $x_2 > 0$; this setting is further investigated in Section (4). Color here depicts the label but is not part of the input.

## A   ADDITIONAL FORMALIZATIONS

### A.1   GENERALIZING THE FRAMEWORK FROM TWO BINARY ATTRIBUTES TO MANY CATEGORICAL ATTRIBUTES

In the most general terms, we consider a setting in which each observation $\mathbf{x} \in \mathcal{X}$ is underlied by $n$ categorical variables $z_1, \ldots, z_n \in \{0, \ldots, C\}$ with $C \in \mathbb{Z}_+$, henceforth *attributes* whose concatenation $\mathbf{z} = (z_1, \ldots, z_n)$ determines the observable input $\mathbf{x}$ via some mapping $g : \mathbb{Z}_{0+}^n \to \mathcal{X}$. We consider the binary classification task of fitting a model $\hat{f} : \mathcal{X} \to \{0, 1\}$ from a given model family $\mathcal{F}$ to predict a binary label for each input. A subset of the attributes in $\mathbf{z}$, without loss of generality $(z_0, \ldots, z_i)$, is taken to define the decision boundary, while the remaining attributes, $z_{i+1}, \ldots, z_n$, are assumed to not be independently predictive of the true classification $y \in \{0, 1\}$. We therefore denote the *discriminant*, $\mathbf{z}_{\text{disc}} = (z_0, \ldots, z_i)$, and the *distractor* $\mathbf{z}_{\text{dist}} = (z_{i+1}, \ldots, z_n)$. For simplicity, we assume that the attributes are binary (*i.e.*, $C = 2$ and $z_i \in \{0, 1\}, \forall i$), and that the discriminant attributes must be jointly active for the classification to change from the null class $y = 0$ (*i.e.*, $y = 1 \iff \mathbf{z}_{\text{disc}} = \mathbf{1}$); the latter simplification allows us to redefine $\mathbf{z}_{\text{disc}} = z_0 \land \cdots \land z_i$ and $\mathbf{z}_{\text{dist}} = z_{i+1} \land \cdots \land z_n$, which is equivalent to the earlier discussion of the illustrative two-attribute case.

### A.2   TRAINING CONDITIONS EXPRESSED IN TERMS OF THE JOINT DISTRIBUTION

We express the training conditions displayed in Fig. (2) and realized in Figure 6 in terms of the joint distribution instead of the parameters $\pi_0, \pi_1$.

1. The cue-conflict condition the upper left and lower right quadrants in Figure 6 and defines the distribution of attributes as

$$\frac{p_{\text{cc}}(\mathbf{z}_{\text{disc}} = 0, \mathbf{z}_{\text{dist}} = 1) = 0.5 \;\middle|\; p_{\text{cc}}(\mathbf{z}_{\text{disc}} = 1, \mathbf{z}_{\text{dist}} = 1) = 0}{p_{\text{cc}}(\mathbf{z}_{\text{disc}} = 0, \mathbf{z}_{\text{dist}} = 0) = 0 \;\middle|\; p_{\text{cc}}(\mathbf{z}_{\text{disc}} = 1, \mathbf{z}_{\text{dist}} = 0) = 0.5 \,.}$$

2. The zero-shot condition populates the bottom left and right quadrants in Figure 6 and defines the distribution of attributes as

$$p_{zs}(\mathbf{z}_{disc} = 0, \mathbf{z}_{dist} = 1) = 0 \quad \Big| \quad p_{zs}(\mathbf{z}_{disc} = 1, \mathbf{z}_{dist} = 1) = 0$$
$$p_{zs}(\mathbf{z}_{disc} = 0, \mathbf{z}_{dist} = 0) = 0.5 \quad \Big| \quad p_{zs}(\mathbf{z}_{disc} = 1, \mathbf{z}_{dist} = 0) = 0.5 \, .$$

3. The partial-exposure condition populates all quadrants but the upper right in Figure 6 and defines the distribution of attributes as

$$p_{pe}(\mathbf{z}_{disc} = 0, \mathbf{z}_{dist} = 1) = 0.25 \quad \Big| \quad p_{pe}(\mathbf{z}_{disc} = 1, \mathbf{z}_{dist} = 1) = 0$$
$$p_{pe}(\mathbf{z}_{disc} = 0, \mathbf{z}_{dist} = 0) = 0.25 \quad \Big| \quad p_{pe}(\mathbf{z}_{disc} = 1, \mathbf{z}_{dist} = 0) = 0.5 \, .$$

## B  MORE CELEBA RESULTS

We include model-specific results, split by ResNet depth and width, in Fig. (**8**). We find no systematic relationship between exemplar- vs. rule-based generalization and depth or width.

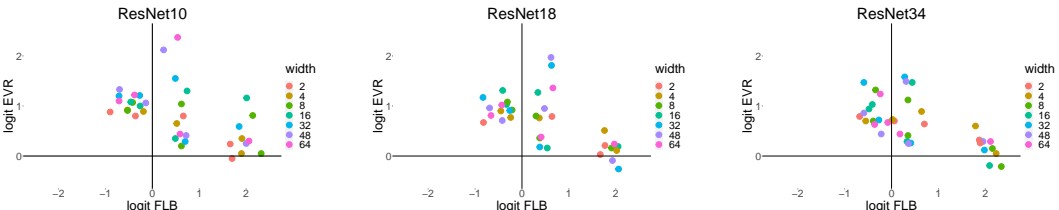

**Figure 8:** CelebA EVR and FLB across feature pairs, averaged across 30 runs, split by depth and width of ResNet.

## C  SPURIOUS CORRELATION UNDERDETERMINES FEATURE DISTRIBUTIONS

The partial-exposure condition ($\pi_0 = 0.5, \pi_1 = 0.0$) in Section (**2**) results in a spurious correlation between the discriminant $\mathbf{z}_{disc}$ and the distractor $\mathbf{z}_{dist}$ ($\rho = 0.58$). To examine behavior in a wider range of data settings, we vary $\pi_0$ and $\pi_1$ as described in Section (**3**), thereby also changing the degree of spurious correlation.

**I. Interpolation towards zero shot.**  We interpolate $\pi_0$ from $0.5$ towards $0.0$, keeping $\pi_1 = 0.0$. This moves us closer to $\pi_0 = \pi_1 = 0.0$, where we have no exposure to $\mathbf{z}_{disc} = 1$ in training. Intuitively, we are reducing the exposure to the new distractor feature value from the partial-exposure condition.

**II. Interpolation to full exposure.**  We interpolate $\pi_1$ from $0.0$ towards $0.5$, keeping $\pi_0 = 0.5$. This moves us closer to $\pi_0 = \pi_1 = 0.5$, where we have equal exposure to all quadrants in training. Here, rather than reducing the exposure to the new distractor feature value, we are equalizing the exposure to it across the discriminant dimension.

**III. Interpolation with matched correlation.**  We report results on this in Sections (**4**) and (**6**). As also depicted in Fig. (**3**), we generate training conditions by changing $\pi_0$ and $\pi_1$ such that we follow a $\rho$-contour away from the partial-exposure condition ($\pi_0 = 0.5, \pi_1 = 0.0, \rho = 0.58$): solid contour in Fig. (**3**). We also match the spurious correlation across the two interpolations in Appendix (**C**)A and B: Fig. (**10**) shows these additional $\rho$-contours as dashed lines.

These different interpolations are depicted in Fig. (**10a**) with different shape/colors.

### C.1  GENERATING INTERPOLATION POINTS

We generate points along all three interpolation lines: from partial exposure towards zero shot ((**C**)I); from partial exposure towards full exposure ((**C**)II); and the equi-correlation line originating from partial exposure ((**C**)III). The interpolating points along each line are selected to balance spurious correlation and feature exposure. In particular, we follow the following procedure:

1. We choose a point that interpolates towards full exposure. We do this by choosing a value of $\pi_1$ between $0.0$ and $0.5$, $\pi^{\text{FE}}$. This gives a data setting, along with a corresponding spurious correlation, $\rho$, computed via Eq. (3):

$$\pi_0 = 0.5 \, ; \qquad \pi_1 = \pi^{\text{FE}} \, ; \qquad \rho = \rho\left(0.5, \pi^{\text{FE}}\right) \, .$$

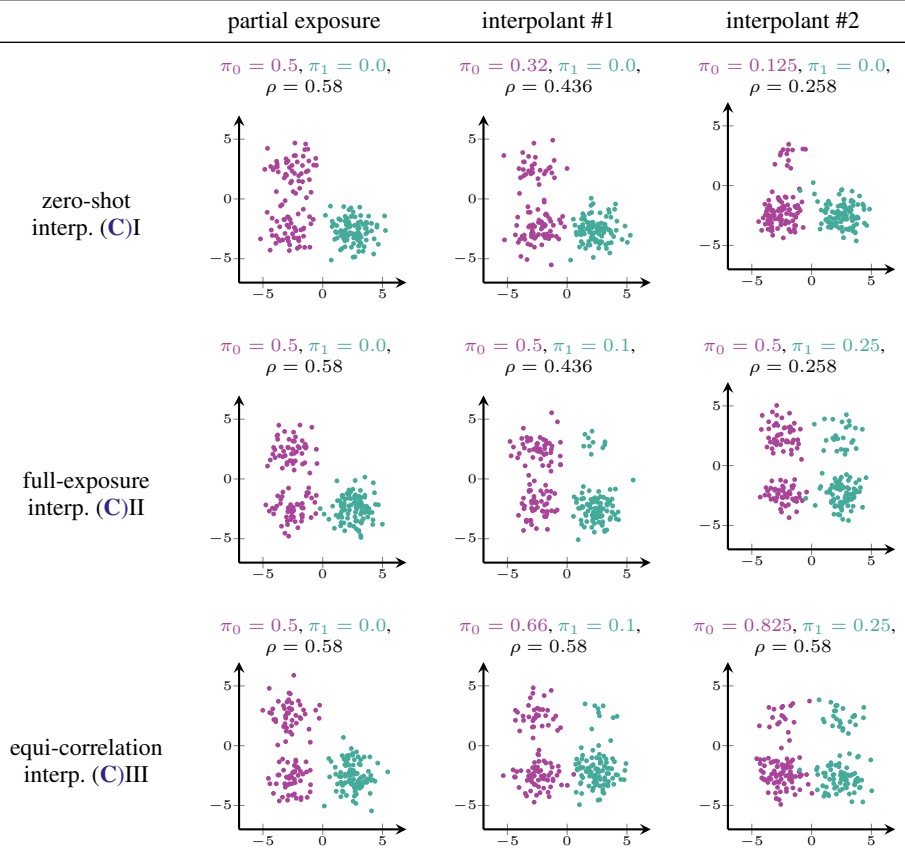

**Figure 9:** We visualize several of the interpolants used for the interpolation analyses.

2. We generate a corresponding point that interpolates towards zero shot. Given the data setting above, we set $\pi_1 = 0.0$ and compute the $\pi_0$ to produce the same $\rho$ as the full-exposure interpolations in Step 1. This gives the data setting:

$$\pi_0 = \pi^{\text{ZS}}\left(\pi^{\text{FE}}\right) \; ; \qquad \pi_1 = 0.0 \; ; \qquad \rho = \rho\left(\pi^{\text{ZS}}\left(\pi^{\text{FE}}\right), 0.0\right) = \rho\left(0.5, \pi^{\text{FE}}\right) \; .$$

3. Finally, we also derive the equi-correlation interpolation from the full-exposure interpolation as follows. We retain $\pi_1$ from the full-exposure condition, but recompute the $\pi_0$ such that the correlation $\rho$ matches the spurious correlation of the pure glspec ($\rho = 0.58$). This gives an additional data setting:

$$\pi_0 = \pi^{\text{EQ}}\left(\pi^{\text{FE}}\right) \; ; \qquad \pi_1 = \pi^{\text{FE}}; \qquad \rho = \rho\left(0.5, 0.0\right) = 0.58 \; .$$

Note that, despite there being three different interpolation lines, the specific interpolants we use are constrained along a single degree of freedom—choosing $\pi^{\text{FE}}$ (Step 1). The data settings for zero shot (Step 2) and equi-correlation (Step 3) are derived from this value.

## C.2 SPECIFIC INTERPOLATION VALUES USED

For all data settings, we generate points along the interpolation lines using the procedure in Appendix (C.1).

For the simple 2D classification setting, we examine two interpolants. In this simple domain, we keep the interpolation distances small, since we expect changes in extrapolation behavior even from small changes.

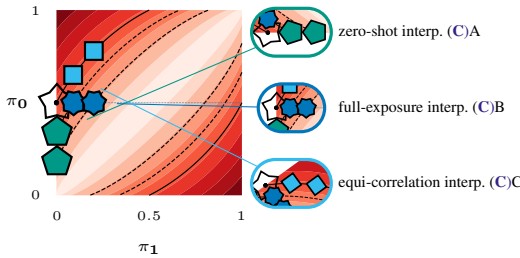
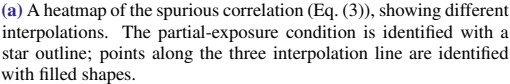

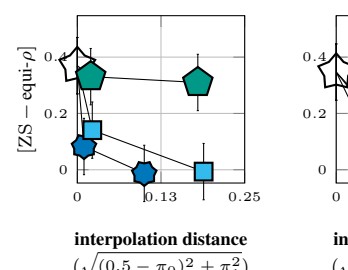

(a) A heatmap of the spurious correlation (Eq. (3)), showing different interpolations. The partial-exposure condition is identified with a star outline; points along the three interpolation line are identified with filled shapes.

(b) Interpolations for NN:16h1d on 2D classification of points-in-a-plane.

(c) Interpolations, ResNet-18-8, CelebA ("wearing lipstick", "mouth open")

**Figure 10:** Interpolations away from the PE: changes in extrapolation behavior under data distribution with the same spurious correlation as in PE, as well as different ways to change spurious correlation.

|  | interpolant 1 | | | interpolant 2 | | |
|---|---|---|---|---|---|---|
|  | $\pi_0$ | $\pi_1$ | $\rho$ | $\pi_0$ | $\pi_1$ | $\rho$ |
| interpolation to zero shot ((**C**)I) | 0.481 | 0.0 | 0.563 | 0.32 | 0.0 | 0.436 |
| interpolation to full exposure ((**C**)II) | 0.5 | 0.01 | 0.563 | 0.5 | 0.1 | 0.436 |
| equi-correlation interpolation ((**C**)III) | 0.519 | 0.01 | 0.58 | 0.661 | 0.1 | 0.58 |

For CelebA, we increase the interpolation distance to reflect the wider range of natural data distributions among feature pairs. The data these interpolation values generate is visualized as the equivalent points-in-a-plane setting in Figure 7.

|  | interpolant 1 | | | interpolant 2 | | |
|---|---|---|---|---|---|---|
|  | $\pi_0$ | $\pi_1$ | $\rho$ | $\pi_0$ | $\pi_1$ | $\rho$ |
| interpolation to zero shot ((**C**)I) | 0.32 | 0.0 | 0.436 | 0.125 | 0.0 | 0.258 |
| interpolation to full exposure ((**C**)II) | 0.5 | 0.1 | 0.436 | 0.5 | 0.25 | 0.258 |
| equi-correlation interpolation ((**C**)III) | 0.66 | 0.1 | 0.58 | 0.825 | 0.25 | 0.58 |

## C.3 INTERPOLATION ANALYSES

### C.3.1 IN THE 2-D CLASSIFICATION EXAMPLE

In the simple setting from Section (4), we vary $\pi_0$, $\pi_1$ for an NN model (NN:16h1d, the NN with lowest ER level overall). Results are in Fig. (10b) and discussed below.

**EVR $\neq$ sensitivity to spurious correlation.** As also discussed in the main text, along the equi-correlation interpolation, the "effective EVR" drops drastically (*i.e.,* the learner generalizes in more rule-based manner) despite no change in spurious correlation.

**Implications for controlling extrapolation.** Despite both having the same $\rho$, interpolating towards full-exposure increases the EVR more than towards zero-shot. This further supports that spurious correlation cannot fully characterize extrapolation behavior. This shows that different *ways* to reduce $\rho$ have different effects on extrapolation, and has important implications for data manipulation methods (*e.g.,* subsampling or augmentation) that aim to directly control this $\rho$.

### C.3.2 IN CELEBA

We see the same effects as in the linear setting: as also discussed in the main text, we see a much smaller gap to the ZS condition despite no change in spurious correlation. We don't find clear effects distinguishing different ways to reduce spurious correlation (interpolation to zero shot ((**C**)I) and interpolation to full exposure ((**C**)II)).

## D    REPRODUCIBILITY DETAILS

We use the criteria from Pineau et al. (2020) omitting the non-applicable *theory* component.

### D.1    MODELS & ALGORITHMS

**An analysis of the complexity (time, space, sample size) of any algorithm.**    The algorithms we employ (LBFGS and stochastic gradient descent on convex and nonconvex problems) are standard, and so we refer the reader to other references to determine their complexities.

### D.2    DATASETS

**The relevant statistics, such as number of examples.**

|                        | 2D             | IMDb               | CelebA[3]                       |
|------------------------|----------------|--------------------|---------------------------------|
| dataset size (train)   | 75             | 21,215             | 4,000 to 40,000                 |
| dataset size (valid)   | 75             | 21,027             | 8,000                           |
| dataset size (test)    | 75             | 13,995             | 20,000                          |
| input space            | $\mathbb{R}^2$ | $\mathbb{R}^{400}$ | $\mathbb{R}^{178\times218\times3}$ |

**The details of train / validation / test splits.**    We do not use a validation set for the simple 2D classification setting and IMDb datasets, but hold out examples for a test set. For CelebA, we follow the authors' division of images in train, validation and test splits.

**An explanation of any data that were excluded, and all pre-processing step.**    As described in the main text, we subsample data to balance attributes within each training condition. For the CelebA domain, we use the following feature pairs to produce the results in Section (6):

| discriminant     | distractor      |
|------------------|-----------------|
| mouth open       | male            |
| wearing lipstick | mouth open      |
| male             | mouth open      |
| male             | high cheekbones |
| male             | blond hair      |
| male             | arched eyebrows |

**A link to a downloadable version of the dataset or simulation environment.**    We use publicly available datasets whose links can be found online.

**For new data collected, a complete description of the data collection process, such as instructions to annotators and methods for quality control.**    We do not collect any new data.

### D.3    CODE

We provide code in a folder of the supplementary entitled `code`.

**Specification of dependencies.**    See `requirements.txt`.

**Training and evaluation code.**    See `main.py`, which imports the module `exposure_bias.train`.

---

[3]The numbers for CelebA are approximate because there are deviations in the availability of images across attribute combinations.

**(Pre-)trained model(s).** We do not provide pre-trained models because of the large number of models evaluated to compute average performances across models.

**README file includes table of results accompanied by precise commands to run to produce those results.** We will clean up and provide Jupyter notebooks to generate all plots in the publicly available code repository after publication.

### D.3.1 EXPERIMENTAL RESULTS

**The range of hyper-parameters considered, method to select the best hyper-parameter configuration, and specification of all hyper-parameters used to generate results.** We use default hyperparameter settings whenever possible; all hyperparameter settings can be found in the accompanying code submission in the files in the folder `exposure-bias/configs/static`.

**The exact number of training and evaluation runs.** For the points-in-a-plane and IMDb settings, we use 20 random seeds, which randomize the model weight initialization. For the CelebA domain, we run 30 seeds for each model configuration, and discard runs that achieve below 75% accuracy on validation set images that belong to the data conditions (quadrants) observed during training.

**A clear definition of the specific measure or statistics used to report results.** We report accuracy as a performance metric on each of the four quadrants depicted in Fig. (2) as well interpolating data settings. We additionally report measures that are a the performance difference between data settings.

**A description of results with central tendency (*e.g.*, mean) & variation (*e.g.*, error bars).** See the main text for a description of results. We include a 95% confidence interval on all reported measures.

**The average runtime for each result, or estimated energy cost.** For the points-in-a-plane, IMDb and CelebA datasets, the average runtime (time to train and evaluate a single model) is 5, 10 and 30 minutes, respectively.

**A description of the computing infrastructure used.** We run experiments serially on an NVIDIA P100 GPU.

