# OpenReview forum: "Distinguishing rule- and exemplar-based generalization in learning systems"
_ICLR.cc/2022/Conference — ICLR 2022 Submitted_

### Official Review · Reviewer_TPBn · 2021-10-27

**Correctness:** 2
**Technical Novelty And Significance:** 2
**Empirical Novelty And Significance:** 2
**Recommendation:** 3
**Confidence:** 4

**Main Review:**

I found that this paper was quite difficult to read since the authors present discussions without clear definitions. Using many acronyms makes the reading difficult too.

For example, “feature-level bias is measured as deviation from chance performance in the CC condition.” When reading CC condition, it says “the data presented in this condition confound color and shape”. What does confound mean here? CC condition is unclear and I do not know feature-level bias.

Similarly, “Exemplar-vs-rule bias is measured by the difference between performance in the PE and ZS conditions—…” When reading PE and ZS, they are not defined either.

Reading on, I do not see formal definitions of feature-level bias exemplar-vs-rule bias. There are measures obtained from models trained by some specific methods. So the proposed approach deals specifically with three types of methods. Why do the authors focus on three types of methods? Why are they representative?

Also, I do not know what spurious correlation means in this paper.

The authors present quite some examples to illustrate their points, but I am confused by their examples. For example, in Figure 1, only two training examples are given. Any prediction is incorrect since there is not enough evidence to learn a model. I cannot see the differences between rule-based and example-based. What rules can we generate based on the two examples?


**Summary Of The Paper:**

This paper studies the extrapolation of machine learning models to unseen regions, and specifically studies two types of biases: feature-level bias (differences in which features are more readily learned) and exemplar-vs-rule bias (differences in how these learned features are used for generalization). Motivated by the studies of exemplar vs. rule-based generalization in cognitive psychology, the authors present a protocol directly probing this trade-off in machine learning systems. The authors present empirical results across a range of models in both expository and real-world image and language domains and demonstrate that using the trad off provides a more complete picture of extrapolation behaviour than existing methods.


**Summary Of The Review:**

The paper is unclear based on the current presentation.

---

> ### Author Response · Authors · 2021-11-20
> **This review does not meet the standards of the review process. (Response to Reviewer TPBn)**
>
> Dear Reviewer TPBn,
>
> We ask that you request the AC to assign another reviewer, as this review does not meet the standards of the review process described in [the ICLR 2022 reviewer guide](https://iclr.cc/Conferences/2022/ReviewerGuide):
> - The review misses definitions of terms and concepts explicitly defined, explained, and visualized in the submission.
> - The review provides no actionable feedback.
> - The abstract of the submission is closely paraphrased in “Summary Of The Paper.”
>
> We respond to specific points of the review below.
>
> Thank you,
>
> The Authors
>
>
> **1. I found that this paper was quite difficult to read since the authors present discussions without clear definitions.**
>
> We note that Reviewer yoH5 states that “the paper is very well-written.” We respond below with clarifications: Much of what the reviewer asks for is already in the paper.
>
> **2. Using many acronyms makes the reading difficult too.**
>
> We introduce 5 acronyms that are clearly defined in the paper:
> - CC is defined on pg. 2 (“Cue conflict (CC, top row, Fig. (1))”) and visualized in Fig. 1.
> - ZS is defined on pg. 3 (“Zero shot (ZS, middle row, Fig. (1))”) and visualized in Fig. 1.
> - PE is defined on pg. 3 (“Partial exposure (PE, bottom row, Fig. (1))”) and visualized in Fig. 1.
> - FLB and EvR are defined on pg. 4 (“Feature-level bias (FLB) and exemplar-vs-rule propensity (EVR) are measured as…”).
>
> The remaining acronyms (GLM, NN, GP) are standard in the field.
>
> The use of acronyms creates visual correspondences between plot markings and the main text. Nevertheless, we are happy to take the feedback and increase clarity surrounding the acronyms. This is a small change that should not take a complete round of conference review.
>
> **3. For example, “feature-level bias is measured as deviation from chance performance in the CC condition.” When reading CC condition, it says “the data presented in this condition confound color and shape”. What does confound mean here? CC condition is unclear and I do not know feature-level bias.**
>
> “[C]onfound color and shape” refers to the fact that color and shape are equally predictive of category boundary, as in the CC condition. CC condition is explained as well as illustrated on pg. 2 and in Fig. 1.
>
> Feature-level bias is defined on pg. 1: “Feature-level bias measures which features a system finds easier or harder to learn. This informs which feature a system will generalize on the basis of when both features are correlated or confounded.”
>
> **4. Similarly, “Exemplar-vs-rule bias is measured by the difference between performance in the PE and ZS conditions—…” When reading PE and ZS, they are not defined either.**
>
> As stated above, PE and ZS are explicitly defined on pgs. 2 & 3.
>
> **5. Reading on, I do not see formal definitions of feature-level bias exemplar-vs-rule bias.**
>
> “Feature-level bias” and “exemplar-vs-rule bias” are described in pg. 1-2 and defined mathematically under “Measuring inductive bias” on pg 4.
>
> **6. There are measures obtained from models trained by some specific methods. So the proposed approach deals specifically with three types of methods. Why do the authors focus on three types of methods? Why are they representative?**
>
> Comparing behavior of a model family across conditions is useful in understanding differences between systems and in revealing the principles underlying systems. This is the basic premise of our work (and of many others); for example, see pg. 2, where we explain “...how differences in classification behavior on this extrapolation isolate feature-level bias as well as exemplar-vs-rule bias.” A similar approach has been used previously, for example, in the literature on the “shape bias” (Ritter 2017a; Hermann 2019; Geirhos 2018), also discussed on pg. 2.
>
> **7. Also, I do not know what spurious correlation means in this paper.**
>
> We refer to the correlation coefficient between distractor and discriminant as the "spurious correlation," as we state on pg. 4: “[t]he correlation coefficient $\rho$ between $\mathbf{z}_\text{disc}$ and $\mathbf{z}_\text{dist}$.” We have not defined the correlation coefficient between two variables mathematically since it is a commonly-known and accepted definition in the field.
>
> **8. The authors present quite some examples to illustrate their points, but I am confused by their examples. For example, in Figure 1, only two training examples are given. Any prediction is incorrect since there is not enough evidence to learn a model … What rules can we generate based on the two examples?**
>
> None of our results are based on training a system in this setting of two examples; this figure is purely for illustration; we realize the visualization as a statistical learning setting, as we state in Sec. 4.
>
> **9. I cannot see the differences between rule-based and example-based.**
>
> The difference is clear in Fig. 1, bottom row, where we demonstrate that the PE condition distinguishes these two modes (columns “rule-based” and “exemplar-based”).

---

> > ### Comment · Reviewer_TPBn · 2021-11-21
> > **Reply to authors**
> >
> > The issues I raise are fundamental in scientific writing. The paper with the issues is not ready for publishing.
> >
> > I use the following example to show my point.
> >
> > My review "For example, “feature-level bias is measured as deviation from chance performance in the CC condition.” When reading CC condition, it says “the data presented in this condition confound color and shape”. What does confound mean here? CC condition is unclear and I do not know feature-level bias.".
> >
> > The authors' reply: “[C]onfound color and shape” refers to the fact that color and shape are equally predictive of category boundary, as in the CC condition. CC condition is explained as well as illustrated on pg. 2 and in Fig. 1.
> >
> > My Re-reply:
> >
> > Feature-level bias is defined by the CC condition. However, the cue conflict (CC) condition has not been defined. Therefore, Feature-level bias is undefined.
> >
> > Let us read the paper again “Cue conflict (CC, top row, Fig. (1)). The data presented in this condition confound color and shape.”
> >
> > Firstly, this is not a definition but an example. A definition should be a statement, such as CC condition is (or means) XXXX.
> >
> > Secondly, this explanation by the example is unclear either since the concept of "confounding" has not been explained. In the authors' reply ""[C]onfound color and shape” refers to the fact that color and shape are equally predictive of category boundary". This sentence is not in the paper. Furthermore, this concept of "confounding" is different from the "confounding" in statistics. A reader with an understanding of statistical confounding will be confused by reading "The data presented in this condition confound color and shape". This further reinforces the point that I made: the authors should define their concepts clearly before using the concepts .

---

> > > ### Author Response · Authors · 2021-11-22
> > > **Second response to follow-up from Reviewer TPBn**
> > >
> > > Our intended meaning of the word "confound" is its dictionary definition. We take the criticism that it might have a more specific meaning in statistics, and have included the explicit sentence "[in the CC condition], color and shape are confounded (i.e., equally predictive of category boundary)" in our revised version to increase clarity. No issue remains here.
> > >
> > > We responded in detail to 9 separate points in the original review to demonstrate that “much of what the reviewer asks for is already in the paper.” This follow-up refers to a single one of these points (which is addressed by the addition of a single phrase, as detailed above) to argue that the paper has issues “fundamental [to] scientific writing.” We see insufficient evidence for this claim in the original review and the follow-up response.
> > >
> > > We are very happy to take and act upon any actionable feedback and suggestions the reviewer might have to improve our submission. At this stage, we feel that this review and the follow-up do not indicate that the submission and our response were "carefully read" and "comprehensively evaluated" as is set out in the [ICLR 2022 Reviewer Guidelines](https://iclr.cc/Conferences/2022/ReviewerGuide), as we stated in the original response.

---

### Official Review · Reviewer_yoH5 · 2021-11-01

**Correctness:** 3
**Technical Novelty And Significance:** 3
**Empirical Novelty And Significance:** 4
**Recommendation:** 6
**Confidence:** 4

**Main Review:**

Strength:
+ Probing the behavior of the learning system by comparing its generalization performance when trained on different datasets is interesting and novel.
+ The synthetic experiment on 2D inputs demonstrate the effectiveness of the measure.
+ The paper is very well-written.

Weakness:
+ Unclear utility.
The main contributions of the paper are the two proposed measures. However, I found it hard to utilize these measures in practical applications.
  + In order to compute the measures, the values for the discriminant features and the distractor features are required. In this situation, there are many existing methods for learning models that are robust against the distractor features [1]. What will be the point of evaluating the bias of a non-robust classifier?
  + For the exemplar-vs-rule propensity (EVR), my understanding is that it captures the extent to which the distractor features are utilized in the learning system. What will be the benefit of EVR compared to evaluating the worst-group accuracy as in [1]? Let’s use CelebA as an example. The training data contains mostly {male, dark_hair}, {female, blond_hair}, {female, dark_hair} images. At test time, [1] evaluates the worst-group accuracy, which is likely to be the performance on {male, blond_hair} (since it is underrepresented in the training data).

Questions and suggestions:
+ How do you interpret the values of the FLB and EVR measures? For instance, in IMDB, the FLB score is -0.3 while the EVR score is 0.54. What do these values mean?
+ I think both measures can be defined in a more generalized setting (where you don’t need to assume the label marginal is uniform).
+ You argued in the paper that the measure is capturing something more than the spurious correlation. I think it will be more precise to state it as **linear** correlation in Figure 3.
+ It will be very interesting to see a contour plot of EVR over $\pi_0$ and $\pi_1$ even in the synthetic experiments.

I am happy to adjust my ratings after the rebuttal period.

[1] Sagawa, Shiori, et al. "Distributionally robust neural networks for group shifts: On the importance of regularization for worst-case generalization." arXiv preprint arXiv:1911.08731 (2019).

**Summary Of The Paper:**

This paper proposes two measures for evaluating the feature-level bias and the exemplar-vs-rule bias in learning systems. Specifically, the authors designed three independent training conditions:
    i. cue conflict: {x1 = 0, x2 = 1, y = 0}, {x1 = 1, x2 = 0, y = 1}
    ii. zero shot: {x1 = 0, x2 = 0, y = 0}, {x1 = 1, x2 = 0, y = 1}
    iii. partial exposure: {x1 = 0, x2 = 0, y = 0}, {x1 = 1, x2 = 0, y = 1}, {x1 = 0, x2 = 1, y = 0}
and one testing condition:
    iv. extrapolation: {x1 = 1, x2 = 1, y = 1}.
The inductive bias of a given learning system is measured by its extrapolation performance difference when trained on different training conditions.

Empirically, the authors first verified their framework on a synthetic dataset with 2D inputs. The results confirm that generalized linear model favors rule-based generalization while Gaussian process favors exemplar-based generalization. On IMDB, the authors show that LSTM models exhibit high feature-level bias (overfitting to the spurious token features) favors exemplar-rule bias. On CelebA, the authors show that ResNet exhibits a wide range of feature-level bias for different features (‘male’ is easier to learn than ‘high cheekbones’). However, across all feature pairs, the model prefers exemplar-based generalization to rule-based generalization.

**Summary Of The Review:**

The authors proposed an interesting approach for measuring the rule- and exemplar-based  generalization for a given learning system. The perspective is novel and the writing is clear. My only concern is the practical utility of the proposed measures.

---

> ### Author Response · Authors · 2021-11-15
> **Author Response to "Official Review of Paper152 by Reviewer yoH5" (1/3)**
>
> We thank the reviewer for their positive comments and valuable feedback. We respond to specific comments below, and hope that these responses more clearly convey the significance of the work. Please let us know if there is more we can answer during the discussion period.
>
> **1. Unclear utility. The main contributions of the paper are the two proposed measures. However, I found it hard to utilize these measures in practical applications.**
>
> We note that the characterizing inductive biases in machine learning models, independent of a downstream application, is of interest to the community. For example: Ritter et al. (ICML 2018) study shape bias; Geirhos et al. (NeurIPS 2018) study inherent robustness to image perturbations; Geirhos et al. (ICLR 2018) study shape vs. texture bias; Rahaman et al. (ICML 2019) and Wang et al. (CVPR 2020) study spectral bias. Based on the existence of numerous concurrent works published at venues like ICLR, we argue that it is sufficient to clearly define, isolate, and study surprising phenomena of models that are in use in current practice, and that not every paper that does so needs to demonstrate improvements in a "practical" application.
>
> Nevertheless, we do have some hypotheses about implications for downstream applications. In the submission, we wrote about implications of our results for systematic generalization (Sec. 7, “Model design for systematic generalization”), data augmentation (Sec. 7, “Data augmentation”), and fairness (Sec. 6, “Practical implications of the EVR” as well as Sec. 8 (the conclusion)). We are happy to add further discussions to the revision if the reviewer feels that specific downstream implications are insufficiently discussed.
>
> **2. In order to compute the measures, the values for the discriminant features and the distractor features are required. In this situation, there are many existing methods for learning models that are robust against the distractor features [1]. What will be the point of evaluating the bias of a non-robust classifier?**
>
> We agree that there exist methods for learning distributionally robust classifiers. However, these methods do not account for the unique contributions of feature-level bias (FLB) and exemplar-vs-rule bias (EvR). DRO in Sagawa et al. (ICLR 2020) focus on reducing scalar spurious correlation, whereas we demonstrate that different ways of manipulating this correlation (by altering $\pi_0$ and $\pi_1$) can have different effects on extrapolation performance, as captured by different mechanisms: feature-level bias (differences in which features are more readily learned) and exemplar-vs-rule bias (differences in how these learned features are used for generalization). Our more granular analysis informs what kinds of data augmentations or model alterations will be most effective in each setting.
>
> More broadly, why do we care about this analysis of non-robust classifiers if we can just make classifiers robust (with DRO or some other method)? We should care about the non-robustness properties of general (non-robust) classifiers because that is what is in use in general, and methods to make classifiers robust commonly make stronger assumptions about the training setting. For example, Sagawa et al. (ICLR 2020) study the application of distributionally robust optimization (DRO) to the group DRO setting, which is related to our 2x2 distractor-discriminant setup; however, their method requires access to subgroup identities to form the training objective (in order to group data in the objective). We consider the more standard training objective with non-subgroup-identified data, and ask the question: Is a particular, commonly-employed model family sensitive to features that are not necessary for classification? We use the subgroup identities in the analysis to answer this question, but the fact that the EvR effect is persistent (across many model families and many feature pairs) suggests that this behavior is general and would persist in settings in which the subgroup identities (spurious features) are not known.
>
> **3. For the exemplar-vs-rule propensity (EVR), my understanding is that it captures the extent to which the distractor features are utilized in the learning system.**
>
> Yes, and to elaborate: EvR measures the extent to which a learning system utilizes an additional feature (ie. the distractor feature) **even when its usage does not aid in optimizing the training objective** (ie. whether it chooses feature-dense ("exemplar-based") over feature-sparse ("rule-based") solutions given no particular data evidence for either).
>
> In contrast, FLB measures how much the distractor feature is used when both distractor and discriminator are equally informative of the category boundary (ie. when they both help equally toward reducing the training objective).

---

> > ### Author Response · Authors · 2021-11-27
> > **Follow-up to Reviewer yoH5**
> >
> > Dear Reviewer yoH5,
> >
> > Since there isn't much time left in the discussion period (ending Nov. 29th), we wanted to check whether you had any remaining concerns that we could address. We believe we addressed the concerns raised in the initial review in our response on Nov 15th, summarized below for convenience. Thank you very much again for taking the time to review our paper and for the feedback!
> >
> > Summary of our response so far: We responded directly to the “utility” concern (Q1). We discussed in more detail the distinctions between our work and prior work (Q2, Q4). We added clarifications in the revision (Q3, Q5, Q6, Q7)—we fixed one detail in Q5 just now (“FLB takes values between -1 and 1” -> “... -0.5 and 0.5”). Regarding Q8, we did not have time to carry out these additional analyses during the brief period that revisions were allowed; since they do not address any particular hypothesis raised in the submission or in the discussion, we are planning it for future work.
> >
> > Thank you,
> >
> > The Authors

---

> > > ### Comment · Reviewer_yoH5 · 2021-11-30
> > > **Thank you for the reply**
> > >
> > > Thank you for the detailed reply. While the practical applicability still seems vague to me, I agree with the authors that it is interesting and important to understand the rule- and exemplar- biases (I have boosted my score).
> > >
> > > *“We consider the more standard training objective with non-subgroup-identified data, and ask the question: Is a particular, commonly-employed model family sensitive to features that are not necessary for classification?”*
> > > + Is it reasonable to assume that a particular model family always exhibit the same level of rule- and exemplar- biases? I guess that will change according to the training data?

---

> > > > ### Author Response · Authors · 2021-12-03
> > > > **Thank you for updating your score!**
> > > >
> > > > We sincerely thank the reviewer for reading our response and increasing their score!
> > > >
> > > > **> Is it reasonable to assume that a particular model family always exhibits the same level of rule- and exemplar- biases? I guess that will change according to the training data?**
> > > >
> > > > We agree that it is important to distinguish something that is strictly a property of a model family from something that is domain-dependent. This is why we evaluate the EvR bias across various settings, demonstrating that it persists in NNs across feature difficulties (i.e., different values of CC), feature values (e.g., various feature combinations in an image classification domain), and domains (e.g., points-in-a-plane; text classification (IMDb); image classification (CelebA)). The evidence that we collected in our empirical studies suggests that NNs are exemplar-based regardless of the setting (which is consistent with prior work demonstrating the ubiquity of sensitivity to spurious correlation). But, as is the nature of empirical work, we cannot guarantee that this holds for *all* domains.
> > > >
> > > > **> the practical applicability still seems vague**
> > > >
> > > > The problem of understanding how NNs extrapolate is important,  timely, and has practical consequences (D'Amour et al., JMLR 2021). Related works published in ICLR and similar conferences that study this question (Johnson et al., CVPR 2017; Sagawa et al., ICLR 2020; Sagawa et al., ICML 2020) use experimental settings similar to ours to manipulate the underlying example/feature distribution of datasets like CelebA. The nature of these careful studies is to understand a phenomenon that has practical importance (NN extrapolation) in a controlled setting. We make analogous (but independent) contributions to understanding NN extrapolation that we believe merit sharing with the ICLR community.
> > > >
> > > > ### References
> > > >
> > > > D'Amour, Alexander, Katherine Heller, Dan Moldovan, Ben Adlam, Babak Alipanahi, Alex Beutel, Christina Chen et al. "Underspecification presents challenges for credibility in modern machine learning." In JMLR, 2021.
> > > >
> > > > Johnson, Justin, Bharath Hariharan, Laurens Van Der Maaten, Li Fei-Fei, C. Lawrence Zitnick, and Ross Girshick. “CLEVR: A diagnostic dataset for compositional language and elementary visual reasoning.” In CVPR, 2017.
> > > >
> > > > Sagawa, Shiori, Pang Wei Koh, Tatsunori B. Hashimoto, and Percy Liang. “Distributionally robust neural networks for group shifts: On the importance of regularization for worst-case generalization.” In ICLR, 2020.
> > > >
> > > > Sagawa, Shiori, Aditi Raghunathan, Pang Wei Koh, and Percy Liang. “An investigation of why overparameterization exacerbates spurious correlations.” In ICML, 2020.

---

> ### Author Response · Authors · 2021-11-15
> **Author Response to "Official Review of Paper152 by Reviewer yoH5" (2/3)**
>
> **4. What will be the benefit of EVR compared to evaluating the worst-group accuracy as in [1]? Let’s use CelebA as an example. The training data contains mostly {male, dark_hair}, {female, blond_hair}, {female, dark_hair} images. At test time, [1] evaluates the worst-group accuracy, which is likely to be the performance on {male, blond_hair} (since it is underrepresented in the training data).**
>
> ​EvR is indeed a function of the *un*represented group’s accuracy (which we always measure as the worst-group test accuracy, as you hypothesize); we call this “accuracy on the held-out/test quadrant.” EvR additionally incorporates a measure of deviation from the performance of the classifier on the extrapolation task **in the absence of linear spurious correlation**: Zero-shot accuracy, which varies significantly on the CelebA task (±10%) as a function of model family and the specific features employed. We additionally measure FLB alongside EvR to demonstrate that they are dependent but not reducible. This allows us to disentangle feature-level bias (FLB) from bias toward feature-dense over feature-sparse solutions (EvR), which are confounded when only measuring worst-group accuracy. These two components (zero-shot baseline and FLB) are not studied in Sagawa et al. (ICLR 2020) and related works.
>
> We also note that our training distribution withholds a quadrant, whereas Sagawa et al. (ICLR 2020) underrepresent a quadrant, as the respective aims of our works are different: We consider the “combinatorial generalization” case of extrapolation to strictly unseen feature combinations, whereas Sagawa et al. (ICLR 2020) consider how reweighting the data distribution to manipulate linear spurious correlation between features (but maintaining the same training support) affects performance on underrepresented data regions. Mixing data from the held-out quadrant into the training data as in Sagawa et al. (ICLR 2020) would complicate measuring (via our EvR measure) the bias for feature-dense vs. feature-sparse solutions because we need to do so **in the absence of any data evidence favoring either more strongly** (which would be a confound).
>
> See for example the effects of manipulating $\pi_0$ and $\pi_1$ in Fig. 6, which demonstrates that fully holding out the “test” quadrant (Fig. 6a) leads to qualitatively different extrapolation behavior compared to settings with the same degree of spurious linear correlation, but where the “test” quadrant is simply underrepresented (Fig. 6b). This is because populating the under-represented quadrant with labeled examples in training provides data evidence for feature-dense vs. feature-sparse solutions.
>
> **5. How do you interpret the values of the FLB and EVR measures? For instance, in IMDB, the FLB score is -0.3 while the EVR score is 0.54. What do these values mean?**
>
> The FLB and EvR measures are meant to compare differences between models in terms of the magnitude of the relevant underlying inductive biases (ie. non-endpoint values are meant to be interpreted as relative measures). The FLB can take values between -1 and 1 (indicating bias toward distractor vs. discriminant features respectively) with 0 representing no feature bias; EvR takes values between 0 and 1 (indicating fully feature-sparse, rule-based extrapolation, and fully feature-dense, exemplar-based extrapolation, respectively). We will clarify this in the revised manuscript.
>
> **6. I think both measures can be defined in a more generalized setting (where you don’t need to assume the label marginal is uniform).**
>
> We agree; however, in this work, we aim to treat the uniform marginal as a control condition, not as an independent variable that is manipulated. This is so we can focus on the independent variables of a) the training conditions (due to how they evidence the inductive biases we study, EvR and FLB), and b) the model family (to see how these inductive biases vary across them). We intend to leave the role of non-uniform label marginals to future work.
>
> **7. You argued in the paper that the measure is capturing something more than the spurious correlation. I think it will be more precise to state it as linear correlation in Figure 3.**
>
> This is correct; we now write: “The linear correlation coefficient $\rho$ between the discriminant and distractor features—henceforth “spurious correlation”—...” We note that Sagawa et al. (ICLR 2020) also use the linear correlation coefficient.
>
> **8. It will be very interesting to see a contour plot of EVR over π0 and π1 even in the synthetic experiments.**
>
> Nice suggestion! We will try to get this done by the 22nd, but it would require many more runs than we performed in the paper for the synthetic setting.

---

> ### Author Response · Authors · 2021-11-15
> **Author Response to "Official Review of Paper152 by Reviewer yoH5" (3/3)**
>
> ### References
>
> [Geirhos, Robert, Carlos R. Medina Temme, Jonas Rauber, Heiko H. Schütt, Matthias Bethge, and Felix A. Wichmann. "Generalisation in humans and deep neural networks." In NeurIPS, 2018.](https://arxiv.org/abs/1808.08750)
>
> [Geirhos, Robert, Patricia Rubisch, Claudio Michaelis, Matthias Bethge, Felix A. Wichmann, and Wieland Brendel. "ImageNet-trained CNNs are biased towards texture; increasing shape bias improves accuracy and robustness." In ICLR, 2018.](https://arxiv.org/abs/1811.12231)
>
> [Rahaman, Nasim, Devansh Arpit, Aristide Baratin, Felix Draxler, Min Lin, Fred A. Hamprecht, Yoshua Bengio, and Aaron C. Courville. "On the Spectral Bias of Deep Neural Networks." In ICML, 2019.](https://arxiv.org/abs/1806.08734)
>
> [Ritter, Samuel, David GT Barrett, Adam Santoro, and Matt M. Botvinick. "Cognitive psychology for deep neural networks: A shape bias case study." In ICML, 2017.](https://arxiv.org/abs/1706.08606)
>
> [Sagawa, Shiori, Pang Wei Koh, Tatsunori B. Hashimoto, and Percy Liang. "Distributionally robust neural networks for group shifts: On the importance of regularization for worst-case generalization." In ICLR, 2020.](https://arxiv.org/abs/1911.08731)
>
> [Wang, Haohan, Xindi Wu, Zeyi Huang, and Eric P. Xing. "High-frequency component helps explain the generalization of convolutional neural networks." In CVPR, 2020.](https://arxiv.org/abs/1905.13545)

---

### Official Review · Reviewer_RJtk · 2021-11-04

**Correctness:** 2
**Technical Novelty And Significance:** 2
**Empirical Novelty And Significance:** 2
**Recommendation:** 3
**Confidence:** 3

**Main Review:**

Some of the ideas introduced by the paper to think about certain inductive biases is interesting. For example, feature-level bias being which features are easier or harder to learn. Here it could be interesting to go into continuous vs. categorical features.

I found some of the exposition unnecessarily confusing. For example, since in Section 2, the color of objects determines their label (i.e. green objects are “dax” and purple objects are “fep”), why use the labels "dax" and "fep" at all rather than just green and purple?

There are many claims that could be substantiated or explained further. For example, why do GPs help with formalizing exemplar-based generalization (Section 4.1)? It would also be clearer to specify which statements have been substantiated through experiments and which have not. For example, Section 4.2 starts off by saying that one NN, one GLM, and one GP has been trained and then make a jump to Figure 4a which shows decision boundaries that are not surprising; I am not sure how it validates the protocol proposed to measure EVR without confounds. Section 4.4 makes a substantial claim ("We also find that different ways to reduce ρ (e.g. by reducing π0 or by increasing π1), give different extrapolation behavior (see Appendix)") based on results pushed to the appendix.

Some sentences could be better worded:
- (Page 6) "EVR increases with exemplar-basedness"
- (Page 7) What is the "held-out quadrant"?


**Summary Of The Paper:**

The paper proposes measures of how two types of explanation methods -- rule based and exemplar based -- generalize and extrapolate to unseen data regions.

**Summary Of The Review:**

The paper is ambitious but falls short in clearly explaining the framework proposed, and substantiating the idea with strong experiments.

---

> ### Author Response · Authors · 2021-11-15
> **Author Response to "Official Review of Paper152 by Reviewer RJtk" (1/2)**
>
> We thank the reviewer for opportunities to improve the clarity of our paper.
>
> **1. The paper is ambitious but falls short in clearly explaining the framework proposed, and substantiating the idea with strong experiments.**
>
> We believe we have addressed the clarity concerns brought up by the reviewer (see below). If any clarity concerns remain, please bring them to our attention during discussion and we will address them in a revision. Otherwise, we sincerely hope that the improvements more clearly convey the contributions of the paper to the reviewer.
>
> We also note that the reviewer has not yet commented on the demonstrations of the exemplar-based behavior of neural network models beyond the synthetic dataset: We have probed this behavior on 2 separate domains (text and vision), and have studied this behavior in depth on the vision domain (with 6 attribute pairs, 18 ResNet depth-width combinations, and 30 random seeds). We believe this is a strong experimental demonstration of the phenomenon introduced in the paper.
>
> **2. Some of the ideas introduced by the paper to think about certain inductive biases is interesting. For example, feature-level bias being which features are easier or harder to learn. Here it could be interesting to go into continuous vs. categorical features.**
>
> We thank the reviewer for their positive assessment of the value of formalizing these inductive biases. We agree that examining continuous features is a natural next step. However, in this paper (which is the first to simultaneously study feature-level bias and exemplar-rule bias), we focused on categorical features to maximize the connection to existing work "combinatorial generalization" (Andreas et al. (CVPR 2017); Johnson et al. (CVPR 2017)) and "subgroup fairness" (Sagawa et al. (ICLR 2020); Sagawa et al. (ICML 2020)). Relatedly, the use of binary multi-label datasets (including specifically CelebA) to explore compositional features is standard in other subfields (Higgins et al. (ICLR 2017); Azadi et al. (IJCV 2020). We will explicitly discuss these papers in the revision.
>
> **3. I found some of the exposition unnecessarily confusing. For example, since in Section 2, the color of objects determines their label (i.e., green objects are "dax" and purple objects are "fep"), why use the labels "dax" and "fep" at all rather than just green and purple?**
>
> The goal of the illustrative example in Fig. 1 was in part to help readers examine their own biases in how they learn novel categories, as we state: "...we encourage the reader to try the experiment themselves to examine their intuitions." We used these arbitrary names to discourage people from using the *a priori* knowledge of what feature is relevant to the underlying category boundary, which is the problem setting used for the ML systems. We will explicitly state this in the revision.
>
> **4. why do GPs help with formalizing exemplar-based generalization (Section 4.1)**
>
> As we state in the manuscript, "[n]on-parametric kernel-based models allow us to formalize exemplar-based generalization." In particular, in the synthetic setting, the GP model as a kernel method operates directly on **similarity in feature space** to training data when extrapolating to new data. The particular kernel we chose (radial basis function) computes similarity over all features, and is thus "feature-dense" and implements exemplar-based reasoning (also see "Exemplar-vs-rule bias" in Sec. 1).
>
> Note that a direct implementation of exemplar-based reasoning is only possible in the synthetic setting, where features over which to compute similarity are known. In contrast, this is not possible in the representation learning contexts (Sec. 5 & 6), where we do not assume knowledge of the discriminant & distractor features *a priori* and thus cannot directly implement a similarity-based kernel classifier. We will clarify this correspondence between kernel-based approaches and exemplar-based generalization in the revised manuscript.
>
> **5. Section 4.2 starts off by saying that one NN, one GLM, and one GP has been trained and then make a jump to Figure 4a which shows decision boundaries that are not surprising**
>
> We emphasize that the synthetic experiment exists to quantitatively confirm intuitions. We chose one kind of model architecture from each broad model family (NN, GLM, GP) and plot decision boundaries over several random initializations in Fig. 4a. We agree that the boundaries reflect expected differences between the models—our key point is that these qualitative differences can be described by the quantitative EvR effect (the GP has high EvR; GLM has low EvR; NN is intermediate). Showing that a model we know to be exemplar-based (the GP) has high EvR and that a model we know to be rule-based (the GLM) has low EvR validates that the EvR behavioral measure indeed captures exemplar-vs-rule generalization. This is discussed in the main text in Sec. 4.2, and we will clarify this motivation in the revised manuscript.

---

> > ### Author Response · Authors · 2021-11-27
> > **Follow-up to Reviewer RJtk**
> >
> > Dear Reviewer RJtk,
> >
> > Since there isn't much time left in the discussion period (ending Nov. 29th), we wanted to check whether you had any remaining concerns that we could address. We believe we addressed the concerns raised in the initial review in our response on Nov 15th, summarized below for convenience. Thank you very much again for taking the time to review our paper and for the feedback!
> >
> > Summary of our response so far: We responded directly to the “clarity” and “experimental strength” concerns (Q1). We clarified that we are “the first to simultaneously study feature-level bias and exemplar-rule bias”, building on the reviewer’s positive comment on the protocol (Q2). We clarified that a central result (insufficiency of spurious linear correlation to predict extrapolation performance) is replicated across multiple domains in the main text, and not simply appearing in the appendix (Q7). We also directly addressed other clarity concerns in a revision (Q3, Q4, Q5, Q6, Q8, Q9).
> >
> > Thank you,
> >
> > The Authors

---

> ### Author Response · Authors · 2021-11-15
> **Author Response to "Official Review of Paper152 by Reviewer RJtk" (2/2)**
>
> **6. I am not sure how it validates the protocol proposed to measure EvR without confounds.**
>
> What we mean here by “without confounds” is that we control for FLB so we can examine EvR in isolation. We will clarify this statement in the revised manuscript.
>
> **7. Section 4.4 makes a substantial claim ("We also find that different ways to reduce ρ (e.g. by reducing π0 or by increasing π1), give different extrapolation behavior (see Appendix)") based on results pushed to the appendix.**
>
> In the main text, we do qualitatively describe results showing that different π0 and π1 values give different extrapolations (first paragraph of Section 4.4), and include quantitative results for another setting of π0 and π1 for the vision domain (Figure 6c), which is discussed in Section 6 under "Controlling spurious correlation." Based on the reviewer’s suggestion, we will briefly discuss the results in the linear case in the main text as well; these results already exist in Figure 10 in the Appendix.
>
> **8. (Page 6) "EVR increases with exemplar-basedness"**
>
> We have rephrased this as follows: “EvR increases with quantities known to increase exemplar-based reasoning.”
>
> **9. (Page 7) What is the "held-out quadrant"?**
>
> As we state under "Training conditions" in Section 3, “[t]he upper-right quadrant in all subfigures of Fig. (2), for which $p(\mathbf{z}_\text{disc} = 1, \mathbf{z}_\text{dist} = 1) = 1$, acts as a *hold-out set* on which we can evaluate generalization to an unseen combination of attribute values … All the analyses in this paper compare model extrapolation to the held-out test quadrant across various training conditions.”
>
>
> ### References
>
> [Azadi, Samaneh, Deepak Pathak, Sayna Ebrahimi, and Trevor Darrell. "Compositional GAN: Learning image-conditional binary composition." In IJCV, 2020.](https://arxiv.org/abs/1807.07560)
>
> [Higgins, Irina, Loic Matthey, Arka Pal, Christopher Burgess, Xavier Glorot, Matthew Botvinick, Shakir Mohamed, and Alexander Lerchner. "Beta-VAE: Learning basic visual concepts with a constrained variational framework." In ICLR, 2017.](https://openreview.net/forum?id=Sy2fzU9gl)
>
> [Johnson, Justin, Bharath Hariharan, Laurens Van Der Maaten, Li Fei-Fei, C. Lawrence Zitnick, and Ross Girshick. "CLEVR: A diagnostic dataset for compositional language and elementary visual reasoning." In CVPR, 2017.](https://arxiv.org/abs/1612.06890)
>
> [Sagawa, Shiori, Pang Wei Koh, Tatsunori B. Hashimoto, and Percy Liang. "Distributionally robust neural networks for group shifts: On the importance of regularization for worst-case generalization." In ICLR, 2020.](https://arxiv.org/abs/1911.08731)
>
> [Sagawa, Shiori, Aditi Raghunathan, Pang Wei Koh, and Percy Liang. "An investigation of why overparameterization exacerbates spurious correlations." In ICML, 2020.](https://arxiv.org/abs/2005.04345)

---

### Official Review · Reviewer_UbRf · 2021-11-04

**Correctness:** 3
**Technical Novelty And Significance:** 2
**Empirical Novelty And Significance:** 2
**Recommendation:** 5
**Confidence:** 1

**Main Review:**

Generalization in the domain of extrapolation, which the authors address in this paper, is one of the critical issues in machine learning. Therefore, they propose a protocol to investigate the problem of inductive bias in extrapolation. Specifically, inspired by psychological research, they propose a protocol to investigate the inductive bias of the learning system towards different features (FLB) and the inductive bias of the learning system towards different ways of using features, either by rule-based or exemplar-based generalization (EVR).

The approach, inspired by psychological research, is interesting. However, we are not convinced that it is a practically valid method, as we have only shown experiments on two datasets.

**Summary Of The Paper:**

The generalization of data distributions to unseen regions, i.e., extrapolation, remains one of the critical challenges in machine learning. The inductive biases of the learner determine such extrapolation. Unfortunately, machine learning systems often do not share the same inductive biases as humans and, as a result, may extrapolate in ways that do not match the analyst's expectations. The authors investigated two different types of such inductive bias: feature-level bias (differences in which features are more easily learned) and exemplar-based and rule-based bias (differences in how learned features are used for generalization). Inspired by these experimental approaches, we have proposed a protocol to investigate this trade-off in learning systems directly. We present empirical results for a range of models and the domains of explanatory images and language. We demonstrate that controlling for feature-level bias while measuring the trade-off between exemplars and rules provides a complete picture of extrapolative behavior than existing formalisms.

**Summary Of The Review:**

The authors' treatment of inductive bias in extrapolation is interesting and will interest many researchers. The proposed protocol, which takes its ideas from psychology, is also interesting. However, we are not convinced that the proposed protocol is practical.

---

> ### Author Response · Authors · 2021-11-19
> **This review does not meet the standards of the review process. (Response to Reviewer UbRf)**
>
> Dear Reviewer UbRf,
>
> We ask that you request the AC to assign another reviewer, as this review does not meet the standards of the review process described in [the ICLR 2022 reviewer guide](https://iclr.cc/Conferences/2022/ReviewerGuide):
> - The review has been marked by the reviewer as extremely low-confidence (“Confidence: 1”).
> - The abstract of the submission is almost exactly paraphrased (twice: once in “Summary Of The Paper” and once in “Main Review”).
> - The review is not substantive: The only critical feedback in the review is that the paper is not “practical” since we have “only shown experiments on two datasets.” We demonstrate results on 1 toy dataset and 2 larger datasets spanning vision and language; this is standard for a conference paper. The reviewer does not indicate what additional datasets/experiments would add value.
>
> Thank you,
>
> The Authors

---

### Decision · Program_Chairs · 2022-01-20

**Decision:**

Reject

**Comment:**

The paper proposes a novel protocol for examining the inductive biases in learning systems, by quantifying the exemplar-rule trade-off (as measured by the exemplar-vs-rule propensity (EVR) defined in Eq. (2)) while controlling for feature-level bias.

Reviewers mostly agree that the problem studied in this paper is practically relevant and that the two bias measures are potentially interesting and (jointly) more informative than existing measures such as spurious correlation. However, a shared concern among the reviewers (with confidences scores >=3) is the clarity of the exposition (e.g., many key concepts such as the data conditions are informally specified [Section 2 (Reviewer TPBn)], some key messages not clearly conveyed in the main paper [Section 3 (Reviewer RJtk)], and results inconclusive or not sufficiently supported by the experimental results [for both the synthetic setting (Reviewer RJtk) and the real-world setting (Reviewer yoH5)]. Based on the above concerns, the reviewers were not convinced that this work is well supported in its current state to merit acceptance for publication.

---

> ### Public Comment · ~Erin_Grant1 · 2022-02-08
> **The meta-review does not reflect the author responses**
>
> Since the meta-review and reviews are persistent and public, we would like to publicly comment on this meta-review. The 3 points raised as grounds for rejection have already been addressed in the author responses (most without meaningful follow-up by reviewers). We are therefore at a loss for how to incorporate feedback from this review process and improve our paper.
>
> > **a shared concern among the reviewers (with confidences scores >=3) is the clarity of the exposition (e.g., many key concepts such as the data conditions are informally specified [Section 2 (Reviewer TPBn)]**
>
> First, the specific claim about data conditions being informally specified is incorrect: Fig. 2 (which is suggestively titled "Formalizing the illustrative experiment") explicitly provides a mathematical specification of each data condition.
>
> Second, we have [responded to concerns about clarity from Reviewer TPBn](https://openreview.net/forum?id=ljCoTzUsdS&noteId=9d5bzuiyAm) with several (9) points which show that "much of what Reviewer TPBn asks for is already in the paper." Neither Reviewer TPBn nor the meta-reviewer has acknowledged all of these points.
>
> Finally, reviewer yoH5 (confidence 4) states that "the paper is very well-written." Therefore, "clarity of the exposition" is simply not "a shared concern among the reviewers (with confidences scores >=3)".
>
> > **some key messages not clearly conveyed in the main paper [Section 3 (Reviewer RJtk)]**
>
> The [specific claim from Reviewer RJtk](https://openreview.net/forum?id=ljCoTzUsdS&noteId=_WCl8DFjdrQ) is that "Section 4.4 makes a substantial claim ("We also find that different ways to reduce ρ (e.g. by reducing π0 or by increasing π1), give different extrapolation behavior (see Appendix)") based on results pushed to the appendix."
>
> [We clarified in the author response](https://openreview.net/forum?id=ljCoTzUsdS&noteId=NykuRS2f7J) that evidence for this claim was displayed in Figure 6c and explicitly discussed in Section 6 under "Controlling spurious correlation” **for the vision domain**. Reviewer RJtk excerpted analogous results for the illustrative **points-in-a-plane** domain, of which detailed figures are indeed in the Appendix due to the page limit.
>
> As such, the key message targeted by Reviewer RJtk was *very clearly conveyed in the main paper at the time of the first submission.* We clarified this in the aforementioned response, but neither Reviewer RJtk nor the meta-reviewer has acknowledged this.
>
> > **results inconclusive or not sufficiently supported by the experimental results [for both the synthetic setting (Reviewer RJtk) and the real-world setting (Reviewer yoH5)]**
>
> This point as written is unclear ("results … not sufficiently supported by … results"), so we assume that the claim is instead that the "conclusions are not supported by the results." (?)
> We directly addressed Reviewer RJtk’s concerns and received zero engagement. Reviewer yoH5 raised their score to above the acceptance threshold after our responses. This leaves us with little actionable feedback on how to actually address the meta-reviewer's comment here in order to revise our paper.
>
> Paper152 Authors